# Probabilistic Decomposed Linear Dynamical Systems for Robust Discovery of Latent Neural Dynamics

**Yenho Chen**[1,3], **Noga Mudrik**[4], **Kyle A. Johnsen**[3],
**Sankaraleengam Alagapan**[2], **Adam S. Charles**[4], **and Christopher J. Rozell**[1,2]

[1]Machine Learning Center, Georgia Institute of Technology
[2]School of Electrical and Computer Engineering, Georgia Institute of Technology
[3]Coulter Dept. of Biomedical Engineering, Emory University and Georgia Institute of Technology
[4]Department of Biomedical Engineering, Mathematical Institute for Data Science,
Center for Imaging Science, Kavli Neuroscience Discovery Institute, Johns Hopkins University
`yenho@gatech.edu, nmudrik1@jhu.edu, kjohnsen@gatech.edu`
`sankar.alagapan@gatech.edu, adamsc@jhu.edu, crozell@gatech.edu`

## Abstract

Time-varying linear state-space models are powerful tools for obtaining mathematically interpretable representations of neural signals. For example, switching and decomposed models describe complex systems using latent variables that evolve according to simple locally linear dynamics. However, existing methods for latent variable estimation are not robust to dynamical noise and system nonlinearity due to noise-sensitive inference procedures and limited model formulations. This can lead to inconsistent results on signals with similar dynamics, limiting the model's ability to provide scientific insight. In this work, we address these limitations and propose a probabilistic approach to latent variable estimation in decomposed models that improves robustness against dynamical noise. Additionally, we introduce an extended latent dynamics model to improve robustness against system nonlinearities. We evaluate our approach on several synthetic dynamical systems, including an empirically-derived brain-computer interface experiment, and demonstrate more accurate latent variable inference in nonlinear systems with diverse noise conditions. Furthermore, we apply our method to a real-world clinical neurophysiology dataset, illustrating the ability to identify interpretable and coherent structure where previous models cannot. [1]

## 1 Introduction

A central goal in computational neuroscience is to develop models capable of discovering latent structure within noisy, high-dimensional neural signals. By identifying hidden relationships within neural recordings, we can begin to understand, predict, and control the behaviors of the underlying systems. Modeling neural time-series is challenging due to the range of temporal dynamics present. For example, there may be gradual short-term fluctuations, abrupt shifts in response to external stimuli, and long-term global drifts resulting from changes in baseline activity levels [36, 43, 39, 11, 7].

Although black-box approaches based on deep learning are available [34, 20, 38], their complexity often obscures the relationships learned from the data, making it difficult to extract scientific insights from these models. As a result, practitioners may favor time-varying linear state-space models which offer mathematically interpretable representations by approximating complex dynamics with simple

---

[1]Code is available at:
`https://github.com/siplab-gt/probabilistic-decomposed-linear-dynamical-systems`

locally linear regimes [26]. However, obtaining latent variable estimates that are robust to dynamical noise and system nonlinearity in these state-space models is challenging. When applied to neural time-series, latent variable estimation may become unstable due to inflexible model formulations or noise-sensitive inference procedures. This can incorrectly produce disparate results for signals generated from the same underlying system.

For example, switching linear dynamical systems (SLDS) and related models [42, 44] segment time-series into discrete linear dynamical states, providing a piecewise linear approximation of the underlying system while highlighting coherent groups of activity. However, the assumption of discrete components can be a poor modeling choice for neural signals that contain continuous-valued fluctuations, such as gradual or random changes to the system speed as seen during neural ramping activity [31] or Lévy walk dynamics in the cerebral cortex [27]. As demonstrated in [30] and our experiments in Section 4.3, when applied to real-world datasets, inference of the switching variables can result in rapid, random oscillations between the discrete modes, indicating that the model is unable to identify meaningful structure in the data.

To address the limitations of discrete states, the decomposed linear dynamical systems (dLDS) model [30] learns a dictionary set of linear dynamical regimes, referred to as dynamic operators (DOs), that can be modified and combined through a linear combination of sparse coefficients. By allowing coefficients to be time-varying and continuous-valued, dLDS naturally captures both gradual changes by adjusting the coefficient magnitudes and abrupt shifts by changing the set of active DOs over time. Inference is accomplished by optimizing over a cost function that encourages data reconstruction while also constraining the structure of the dynamic coefficients to be sparse and temporally smooth.

Unfortunately, there are two critical shortcomings that prevent the robust inference of the latent variables in dLDS. First, the cost-based inference procedure is sensitive to noise, because of the regularization term encouraging temporal smoothness. This term sequentially propagates errors from noisy coefficient estimates over the length of the time series. Consequently, the model may produce inconsistent coefficient estimates on similar signals and have poor multi-step inference performance, indicating that the learned dynamics are unable to generalize well beyond a single time step. Second, the original latent dynamics model lacks a method for accurately representing systems with multiple fixed points, causing DO coefficients to oscillate or switch arbitrarily in a way that may not align with the underlying process. To learn an effective decomposed model in practice, we require a strategy that provides robust estimates of latent variables despite the presence of noise and system nonlinearity.

In this work, we address these limitations by introducing the probabilistic decomposed linear dynamical systems (p-dLDS) model. Our approach improves robustness of latent variable estimation while maintaining the richness of a decomposed dynamical systems model. First, we propose a probabilistic inference procedure that reduces the model's sensitivity to temporal noise by accounting for uncertainty in the latent variable estimates over time. Namely, we introduce time-informed hierarchical variables that encourage both sparse and smooth model coefficients. We devise a variational expectation maximization (vEM) procedure to perform inference and learning over this probabilistic structure. Second, we incorporate a time-varying offset term to model systems that orbit multiple fixed points. While we analytically identify model degeneracies with this formulation, we propose an additive decomposition strategy that prevents convergence to trivial solutions.

Through several synthetic examples, we demonstrate how these contributions lead to improved accuracy and robustness of latent variable estimation despite difficult noise conditions. We extend these results to an empirically-derived brain-computer interface experiment [8], showcasing robustness to highly nonlinear observation functions and the ability to extract meaningful insights from the learned latent variables. Finally, we illustrate how our method effectively identifies interpretable and coherent structure in a clinical neurophysiology dataset where previous models are unsuccessful.

## 2  Background and Related Work

**State Space Models.** Our goal is to accurately describe the evolution of high-dimensional time-series data $\boldsymbol{y}_t \in \mathbb{R}^M$ with the following state-space equations,

$$\boldsymbol{y}_t = \boldsymbol{D}\boldsymbol{x}_t + \boldsymbol{d} + \boldsymbol{\epsilon}_{y_t}, \quad \boldsymbol{\epsilon}_{y_t} \sim \mathcal{N}(0, \boldsymbol{\Sigma}_y), \quad \text{(Observations)} \tag{1}$$
$$\boldsymbol{x}_t = f_t(\boldsymbol{x}_{t-1}) + \boldsymbol{\epsilon}_{x_t}, \quad \boldsymbol{\epsilon}_{x_t} \sim \mathcal{N}(0, \boldsymbol{\Sigma}_x), \qquad \text{(Dynamics)}$$

where $\boldsymbol{x}_t \in \mathbb{R}^N$ is the latent state, $f_t(\cdot)$ is the dynamics function, and $\boldsymbol{D} \in \mathbb{R}^{M \times N}$ and $\boldsymbol{d} \in \mathbb{R}^M$ describe a linear observation function. Our work focuses on the case when $N < M$, which compresses high-dimensional signals into a low-dimensional latent space. By choosing $f_t$ to be a time-varying linear operator, we can approximate complex nonlinear dynamics with simple locally-linear components, balancing expressivity with mathematical interpretability. However, learning a time-varying linear operator from data can be challenging, and typically requires additional constraints on the underlying generative model to identify meaningful representations.

**Switching Linear Dynamical System (SLDS).** SLDS approximates nonlinear systems by introducing a discrete switching variable $z_t = \{1, \ldots, K\}$ into the time-varying linear dynamics equation,

$$\boldsymbol{x}_{t+1} = \boldsymbol{x}_t + \boldsymbol{F}_{z_t}\boldsymbol{x}_t + \boldsymbol{b}_{z_t} + \boldsymbol{\epsilon}_{x_t}.$$

At each time step $t$, the latent state $\boldsymbol{x}_t$ evolves according to the $z_t$-th linear regime defined by $\boldsymbol{F}_{z_t} \in \mathbb{R}^{N \times N}$ and $\boldsymbol{b}_{z_t} \in \mathbb{R}^N$ while the switching variables evolve according to a Markov matrix. Inference is performed through a vEM algorithm, where the approximate posterior of the latent variables is estimated through coordinate ascent updates over tractable subgraphs. There are many extensions of SLDS, such as rSLDS [26] which modifies its generative behavior by informing the transitions of $z_t$ with $x_{t-1}$. However, switching models are inherently limited when describing complex signals due to their discrete formulation. For instance, a switched representation is unable to learn that a dynamic regime may exhibit a range of variations. In neural systems, these variations may arise from random spiking processes [37, 40] or systems with randomly distributed speeds [41, 12]. In SLDS, each variation is learned as a separate discrete state, thus obscuring that the learned states are related. Furthermore, the switching formulation cannot adapt the learned system to unseen variations (i.e. different levels of random speeds). This can produce unstable inference behavior, where the switching state oscillates unpredictably or collapses to a single uninformative state.

**Decomposed Linear Dynamical Systems (dLDS).** dLDS [30] relaxes the discrete formulation by approximating nonlinear and nonstationary signals with a time-varying mixture of linear dynamical systems (LDS) defined by the following equations and constraints,

$$\boldsymbol{x}_{t+1} = \boldsymbol{x}_t + \boldsymbol{F}_t\boldsymbol{x}_t + \boldsymbol{\epsilon}_{x_t}, \qquad \boldsymbol{F}_t = \sum_{k=1}^K \boldsymbol{f}_k c_{t,k}, \qquad \text{s.t. } \boldsymbol{c}_t \text{ is sparse.} \qquad (2)$$

Every transition $\boldsymbol{F}_t$ is decomposed as a linear combination of sparse coefficients $\boldsymbol{c}_t \in \mathbb{R}^K$ and a dictionary of $K$ DOs $\boldsymbol{f}_k \in \mathbb{R}^{N \times N}$. Figure 1A shows the corresponding graphical model. Inference of the latent variables is accomplished by solving the Basis Pursuit Denoising with Dynamic Filtering (BPDN-DF) [6] objective sequentially for all $t$ and $\lambda_0, \lambda_1, \lambda_2 > 0$,

$$\widehat{\boldsymbol{x}}_t, \widehat{\boldsymbol{c}}_t = \underset{\boldsymbol{x}_t, \boldsymbol{c}_t}{\arg\min} \|\boldsymbol{y}_t - \boldsymbol{D}\boldsymbol{x}_t\|_2^2 + \lambda_0 \|\boldsymbol{x}_t - \widehat{\boldsymbol{x}}_{t-1} - \boldsymbol{F}_t\widehat{\boldsymbol{x}}_{t-1}\|_2^2 + \lambda_1 \|\boldsymbol{c}_t\|_1 + \lambda_2 \|\boldsymbol{c}_t - \widehat{\boldsymbol{c}}_{t-1}\|_2^2.$$

This produces a point estimate of $\boldsymbol{x}_t$ and $\boldsymbol{c}_t$ that matches the likelihood function resulting from Equations (1) and (2). In this objective, the dynamic coefficients are encouraged to be sparse through the $\ell_1$ penalty and temporally smooth through the $\ell_2$ penalty centered around the previous coefficient estimate. However, this approach is sensitive to noise because inference relies on propagating noisy point estimates of $\widehat{\boldsymbol{c}}_{t-1}$ over time. As a result, BPDN-DF may accumulate errors that can lead to significantly different coefficient estimates on signals sampled from the same generative process. Furthermore, the lack of robustness to noise can degrade multi-step inference performance, causing the inferred system to quickly diverge from the true system. This suggests that the inferred latent variables only capture the local activity narrowly and are unable to accurately represent the dynamics beyond a single time-step. Another drawback of dLDS arises from the dynamics model in equation (2) which implicitly assumes that the observed dynamics contain a single fixed point that revolves around the origin. This limits dLDS's ability to model systems that cannot be easily mean-centered such as those with multiple fixed points or nonstationary drifts.

**Sparse Bayesian Learning with Dynamic Filtering.** Sparsity is achieved in probabilistic models through hierarchical scale-mixture priors [10, 2, 4]. To integrate dynamical information into probabilistic sparse signal inference, previous work [33] proposes the Sparse Bayesian Learning with Dynamic Filtering (SBL-DF) framework where the following hierarchical model is defined,

$$p(\mathbf{x}_t, \mathbf{c}_t, \gamma_t) = p(\mathbf{x}_t|\mathbf{c}_t) \prod_{k=1}^K p(c_{t,k}|\gamma_{t,k})p(\gamma_{t,k}|a_{t,k}, b_{t,k}). \qquad (3)$$

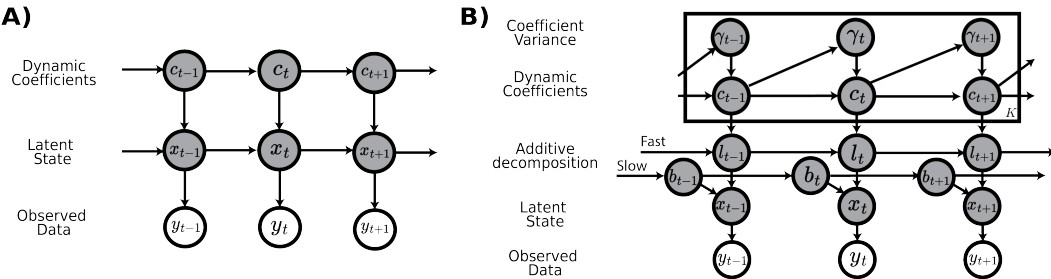

Figure 1: **(A)** Graphical model of dLDS. **(B)** p-dLDS includes hierarchical variables for probabilistic sparse inference and reparameterizes the latent space to include a time-varying offset term.

Here, $p(\mathbf{x}_t|\mathbf{c}_t) = \mathcal{N}(\mathbf{\Phi}\mathbf{c}_t, \lambda_t\mathbf{I})$ specifies the likelihood, where $\mathbf{\Phi}$ is a measurement matrix. Sparsity is encouraged through the zero-mean Gaussian priors $p(c_{t,k}|\gamma_{t,k}) = \mathcal{N}(0, \gamma_{t,k})$ independently placed on each element of the sparse vector. The variance parameters $\gamma_{t,k}$ are defined by an inverse gamma hyperprior $p(\gamma_{t,k}|\alpha_{t,k}, \beta_{t,k}) = \mathcal{IG}_{\gamma_{t,k}}(\alpha_{t,k}, \beta_{t,k})$ with shape parameters $\alpha_{t,k}$ and $\beta_{t,k}$. When marginalizing over $\gamma_{t,k}$'s, we see that $p(c_{t,k}|\alpha_{t,k}, \beta_{t,k})$ becomes a t-distribution known for its high kurtosis, which is essential for producing sparse solutions. To propogate dynamics information, the past estimate $\widehat{c}_{t-1}$ informs the hyperprior parameters of the next estimate such that $b_{t,k}/a_{t,k} = c_{t-1,k}^2$.

# 3 Probabilistic Decomposed Linear Dynamical Systems

We build upon dLDS and propose a probabilistic decomposed linear dynamical systems (p-dLDS) model. Rather than propagating noisy point estimates of the latent variables during inference, we improve robustness by marginalizing over uncertainty with respect to time. Additionally, we propose a tractable method for extending the dynamics model to systems with multiple fixed points.

## 3.1 Time-varying offset term

Note that for any parameter setting, the dLDS local dynamics (eq. (2)) reduces to a linear dynamical system (LDS) that is characterized by a single fixed point centered around the origin. Yet, real-world dynamical systems often consist of much more complicated behaviors. Nonlinearities can cause a signal to navigate through multiple fixed points throughout its trajectory, while nonstationarities may change the behavior of the system entirely with new fixed points emerging or disappearing. Simple preprocessing measures, such as mean-centering the data, are inadequate to account for these behaviors. To enable robustness against these behaviors, we introduce a time-varying offset term $\boldsymbol{b}_t \in \mathbb{R}^N$ into Equation (2) as a flexible way to account for dynamics not readily captured by the original dLDS latent dynamics model.

**Lemma 1.** *Let the transition between any two state vectors $\boldsymbol{x}_t, \boldsymbol{x}_{t+1} \in \mathbb{R}^N$ be defined by the linear dynamics matrix $\boldsymbol{F}_t \in \mathbb{R}^{N \times N}$ and the dynamics offset $\boldsymbol{b}_t \in \mathbb{R}^N$. For any $\lambda > 0$, the objective,*

$$\underset{\boldsymbol{F}_t, \boldsymbol{b}_t}{\arg\min} \|\boldsymbol{x}_{t+1} - \boldsymbol{x}_t - \boldsymbol{F}_t\boldsymbol{x}_t - \boldsymbol{b}_t\|_2^2 + \lambda\|\boldsymbol{F}_t\|_2^2,$$

*is minimized when $\boldsymbol{F}_t = \boldsymbol{0}$ and $\boldsymbol{b}_t = \boldsymbol{x}_{t+1} - \boldsymbol{x}_t$.*

This result, proven in Appendix B.1, reveals that introducing a time-varying offset term makes inference of the dynamics a degenerate problem. While the solution in Lemma 1 minimizes the objective, it fails to capture any meaningful structure in $\boldsymbol{F}_t$ as the result of $\boldsymbol{b}_t$ being unconstrained. To prevent the convergence to these trivial solutions, we decompose the latent state space as,

$$\boldsymbol{x}_t = \boldsymbol{l}_t + \boldsymbol{b}_t, \tag{4}$$

where $\boldsymbol{l}_t$ captures fast dynamics and $\boldsymbol{b}_t$ captures slow-varying trend behavior. These latent variables follow the dynamic equations $\boldsymbol{l}_{t+1} = \boldsymbol{l}_t + \boldsymbol{F}_t\boldsymbol{l}_t + \boldsymbol{\epsilon}_{l_{t+1}}$ and $\boldsymbol{b}_{t+1} = \boldsymbol{b}_t + \boldsymbol{\epsilon}_{b_{t+1}}$ where $\boldsymbol{\epsilon}_l, \boldsymbol{\epsilon}_b \in \mathbb{R}^N$ represent noise sampled from $\boldsymbol{\epsilon}_b \sim \mathcal{N}(0, \boldsymbol{\Sigma}_b)$ and $\boldsymbol{\epsilon}_l \sim \mathcal{N}(0, \boldsymbol{\Sigma}_l)$ respectively.

## 3.2 Probabilistic Time-Informed Sparsity

In decomposed models, we aim to achieve two goals simultaneously: sparsity and smoothness of coefficients over time. Motivated by this, we incorporate dynamics-informed probabilistic structure. First, we assume that each coefficient evolves independently of the others. Second, we introduce a hierarchical variance parameter $\gamma_{t,k}$ that controls the sparsity for each $c_{t,k}$. Moreover, we introduce dynamics information during sparse inference by encouraging a similar active support set in consecutive time slices through the variance hyperpriors. Put together, the resulting coefficient transition density in p-dLDS becomes,

$$p(\boldsymbol{c}_t, \boldsymbol{\gamma}_t | \boldsymbol{c}_{t-1}) := p(\boldsymbol{c}_t | \boldsymbol{c}_{t-1}, \boldsymbol{\gamma}_t) p(\boldsymbol{\gamma}_t | \boldsymbol{c}_{t-1}) = \prod_{k=1}^{K} p(c_{t,k} | c_{t-1,k} \gamma_{t,k}) p(\gamma_{t,k} | c_{t-1,k}). \tag{5}$$

We define the first term on the right-hand side with the following functional form,

$$p(c_{t,k} | c_{t-1,k}, \gamma_t) \propto \exp\left( -\frac{c_{t,k}^2}{2\gamma_{t,k}} - \frac{(c_{t,k} - c_{t-1,k})^2}{2\sigma_{t-1,k}^2} \right) \propto \mathcal{N}(c_{t-1,k}, \sigma_{t-1,k}^2) \mathcal{N}(0, \gamma_{t,k}). \tag{6}$$

This density captures the constraints of sparsity and smoothness for the inferred coefficients $c_{t,k}$. When the variance around zero $\gamma_{t,k}$ is small, this structure promotes sparsity by shrinking coefficient values towards zero. Conversely, when the variance around the previous time step $\sigma_{t-1,k}^2$ is small, it encourages smoooothness by shrinking coefficients towards the previous value. While the idea of combining two shrinkage effects in a single density has been explored in previous works [15, 5, 24, 18], those approaches generally require manual balancing of the two penalties. In contrast, we devise a procedure in the following section that estimates these variance parameters automatically during inference and learning.

The second density on the right-hand side from equation (5) is defined similarly to the hyperprior in SBL-DF. (i.e., $p(\gamma_{t,k} | c_{t-1,k}) = \mathcal{IG}(\xi, \xi c_{t-1,k}^2)$ where $\xi$ weighs the influence of the dynamics when estimating $\gamma_{t,k}$). The resulting graphical model is shown in Figure 1B. We note that since the value of the previous coefficient is squared, the overall prior placed on the inverse gamma density follows a $\chi^2$ distribution.

## 3.3 Inference and Learning

The joint distribution of p-dLDS is given by,

$$p(\boldsymbol{x}, \boldsymbol{y}, \boldsymbol{c}, \boldsymbol{\gamma} | \theta) = p(\boldsymbol{x}_1) \left[ \prod_{t=1}^{T} p(\boldsymbol{y}_t | \boldsymbol{x}_t) \right]$$
$$\left[ \prod_{t=1}^{T-1} p(\boldsymbol{x}_{t+1} | \boldsymbol{x}_t, \boldsymbol{c}_t) \left[ \prod_{k=1}^{K} p(c_{t+1,k} | c_{t,k}, \gamma_{t+1,k}) p(\gamma_{t+1,k} | c_{t,k}) \right] \right], \tag{7}$$

where we denote $\boldsymbol{x} = \boldsymbol{x}_{1:T}$ for brevity. Exact posterior inference is intractable due to the nonconjugacy introduced by incorporating time-informed sparsity-inducing structure into the graphical model. As a result, we devise a variational expectation maximization (vEM) procedure where the approximate posterior is factorized as

$$p(\boldsymbol{x}, \boldsymbol{c}, \boldsymbol{\gamma} | \boldsymbol{y}, \theta) \approx q(\boldsymbol{x}) q(\boldsymbol{c}, \boldsymbol{\gamma}).$$

Here, the parameters are given by $\theta \in \{\boldsymbol{f}_{1:K}, \boldsymbol{D}, \boldsymbol{d}, \boldsymbol{\Sigma}_y, \boldsymbol{\Sigma}_x, \boldsymbol{\Sigma}_c\}$. Our approach contrasts with BPDN-DF, which estimates latent variables through separate $\ell_1$ problems at each point in time. Instead, we preserve the time-dependence structure within each class of latent variables and leverage efficient inference algorithms that marginalize over uncertainty with respect to time. In general, we seek to maximize the variational lower bound,

$$\mathcal{L}_q(\theta) = \mathbb{E}_{q(\boldsymbol{x})q(\boldsymbol{c},\boldsymbol{\gamma})}[\log p(\boldsymbol{y}, \boldsymbol{x}, \boldsymbol{c}, \boldsymbol{\gamma} | \theta) - \log q(\boldsymbol{x}) q(\boldsymbol{c}, \boldsymbol{\gamma})],$$

with coordinate ascent updates on the latent state posterior, the dynamics coefficients posterior, and the model parameters.

**Updating Latent State Posterior.** The optimal coordinate ascent variational update is given by,

$$q(\boldsymbol{x}) \propto \exp\left(\mathbb{E}_{q(\boldsymbol{c}, \boldsymbol{\gamma})}\left[\log p(\boldsymbol{x}, \boldsymbol{y}, \boldsymbol{c}, \boldsymbol{\gamma}|\theta)\right]\right). \tag{8}$$

Our assumed decomposition in equation (4) allows us to define the latent state transition density as $p(\boldsymbol{x}_{t+1}|\boldsymbol{x}_t, \boldsymbol{c}_t) = p(\boldsymbol{x}_{t+1} = \boldsymbol{l}_{t+1} + \boldsymbol{b}_{t+1}) = \mathcal{N}(\boldsymbol{x}_{t+1}; \boldsymbol{l}_t + \boldsymbol{F}_t \boldsymbol{l}_t + \boldsymbol{b}_t)$. Substituting this into equations (7) and (8), we get that the optimal coordinate ascent approximate posterior becomes,

$$q(\boldsymbol{x}) = \mathcal{N}(\boldsymbol{x}_1; \boldsymbol{\mu}_1, \Sigma_1) \left[\prod_{t=2}^{T} \mathcal{N}(\boldsymbol{y}_t; \boldsymbol{D}\boldsymbol{x}_t + \boldsymbol{d}, \boldsymbol{\Sigma}_y)\right] \left[\prod_{t=1}^{T} \mathcal{N}(\boldsymbol{x}_t; \boldsymbol{l}_{t-1} + \boldsymbol{F}_{t-1}\boldsymbol{l}_{t-1} + \boldsymbol{b}_t, \boldsymbol{\Sigma}_x)\right], \tag{9}$$

where, $\boldsymbol{\mu}_1$ and $\boldsymbol{\Sigma}_1$ are the mean and covariance of the initial state.

**Lemma 2.** *Let $\boldsymbol{l}, \boldsymbol{b} \in \mathbb{R}^N$ be independent random variables such that $\boldsymbol{l} \sim p(\boldsymbol{l})$ and $\boldsymbol{b} \sim p(\boldsymbol{b})$. Their sum $\boldsymbol{x} = \boldsymbol{l} + \boldsymbol{b}$ is distributed according to $\mathcal{N}(\boldsymbol{\mu}_l + \boldsymbol{\mu}_b, \boldsymbol{\Sigma}_l + \boldsymbol{\Sigma}_b)$ when 1) $p(\boldsymbol{b}) = \mathcal{N}(\boldsymbol{\mu}_b, \boldsymbol{\Sigma}_b)$ and $p(\boldsymbol{l}) = \mathcal{N}(\boldsymbol{\mu}_l, \boldsymbol{\Sigma}_l)$ and when 2) $p(\boldsymbol{b}) = \delta(\boldsymbol{b} - \boldsymbol{\mu}_b)$ and $p(\boldsymbol{l}) = \mathcal{N}(\boldsymbol{\mu}_l, \boldsymbol{\Sigma}_l + \boldsymbol{\Sigma}_b)$.*

We leverage Lemma 2 to reparameterize the state space trajectories into a deterministic and a stochastic component. The deterministic component captures the slow-moving offset density $q(\boldsymbol{b}) = \prod_{t=1}^{T} \delta(\boldsymbol{b}_t - \widehat{\boldsymbol{b}}_t)$, where $\widehat{\boldsymbol{b}}_t$ is estimated using a moving average of window size $S$ which can be efficiently parallelized. The remaining dynamics are captured in the stochastic component. We define the family of variational distributions to be the class of linear Gaussian state space models, such that $q(\boldsymbol{l}) = \mathcal{N}(\boldsymbol{l}_1, \boldsymbol{\Sigma}_1) \prod_{t=2}^{T} \mathcal{N}(\boldsymbol{l}_{t-1} + \boldsymbol{F}_{t-1}\boldsymbol{l}_{t-1}, \boldsymbol{\Sigma}_x)$. Conditioned on estimates $\widehat{\boldsymbol{b}}$ and samples of $\boldsymbol{c}$, the optimal coordinate ascent variational update for $q(\boldsymbol{l})$ is efficiently computed using the Kalman Smoother [35]. We provide a full derivation of the update rule and Lemma 2 in Appendix B.2.

**Updating Dynamics Coefficient Posterior.** Sparse probabilistic representations introduce non-Gaussian factors which prevent closed-form message passing inference. Specifically, nonconjugacy arises from the inverse gamma term $p(\gamma_{t+1,k}|c_{t,k})$ since it is parameterized by $c_{t,k}^2 \sim \chi^2$. Moreover, the posterior distribution over the coefficients is highly multi-modal as a result of the implicit t-distribution in our hierarchical model. To update the coefficient posteriors, we propose a three-step procedure, where we factorize $q(\boldsymbol{c}, \boldsymbol{\gamma}) = q(\boldsymbol{c})q(\boldsymbol{\gamma})$. First, we obtain an initial estimate of the variational distributions using SBL-DF. Second, we update $q(c_{t,k}) = \mathcal{N}(c_{t,k}^*, \widehat{\gamma}_{t,k})$ using stochastic gradient descent (SGD) over

$$\boldsymbol{c}^* = \arg\max_{\boldsymbol{c}} \sum_{t=1}^{T-1} \log p(\widehat{\boldsymbol{l}}_{t+1}|\widehat{\boldsymbol{l}}_t, \boldsymbol{c}_t) + \log p(\boldsymbol{c}_{t+1}|\boldsymbol{c}_t, \widehat{\gamma}_{t+1}) + \log p(\widehat{\gamma}_{t+1}|\boldsymbol{c}_t), \tag{10}$$

where we have estimated the expectation in the optimal coordinate ascent update rule using samples from $\widehat{\gamma} \sim q(\boldsymbol{\gamma})$ and $\widehat{\boldsymbol{l}} \sim q(\boldsymbol{l})$. To retain coefficient sparsity, we only update coefficients within the active support set. In our work, this is defined as coefficients that have an initial estimate of $|c_{t,k}| > \eta$ where $\eta = 10^{-4}$. Finally, we update $q(\boldsymbol{\gamma})$ based on closed form conjugacy rules.

**Update Parameters.** Given our updated posteriors of the latent variables, we proceed to update the model parameters based on the ELBO,

$$\theta^* = \arg\max_{\theta} \mathbb{E}_{q(\boldsymbol{x})q(\boldsymbol{c})q(\boldsymbol{\gamma})}\left[p(\boldsymbol{y}, \boldsymbol{c}, \boldsymbol{x}, \boldsymbol{\gamma}|\theta) - \log q(\boldsymbol{x})q(\boldsymbol{c})q(\boldsymbol{\gamma})\right] \approx \arg\max_{\theta} p(\boldsymbol{y}, \widehat{\boldsymbol{c}}, \widehat{\boldsymbol{x}}, \widehat{\boldsymbol{\gamma}}|\theta), \tag{11}$$

where we estimate the expectation with samples from our variational distributions and drop terms not dependent on $\theta$. We use SGD to update all model parameters, which is possible when we assume that the covariance matrices have diagonal structure.

## 4   Results

We demonstrate p-dLDS in a variety of synthetic examples, highlighting improved robustness to noise and system nonlinearity. Additionally, we apply our model to a clinical neurophysiology dataset, revealing interpretable patterns where previous methods fail. We compare our method against SLDS, rSLDS, and dLDS as described in Section 2. All datasets are split 50:50 for training and testing. Due to space constraints, we provide full descriptions of the simulation setup in Appendix C and metric definitions in Appendix D.

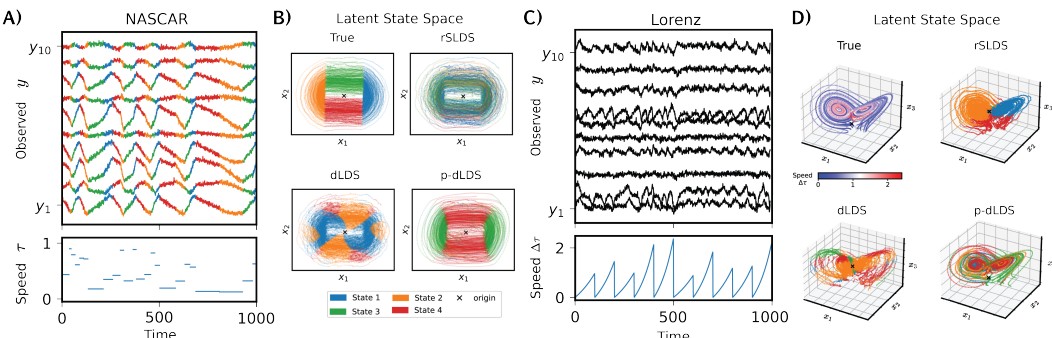

Figure 2: **Probabilistic model and offset term reduce estimation errors. (A)** Example trial from the NASCAR experiment colored by the true switching labels (not provided during training). Each track segment has a random speed $\tau$. **(B)** Inferred state space, colored by discrete state or dominant coefficients. p-dLDS identifies correct track segments. **(C)** Example trial from the Lorenz experiment. The speed ramps according to the time intervals $\Delta\tau$ in an ODE solver. **(D)** Inferred state space, colored by the dominant coefficients. The time-varying offset term allows p-dLDS coefficients to switch according to the true speed and accurately model the two fixed points in the opposing lobes.

Table 1: Metrics for synthetic dynamical systems. Bold means best performance. ($\uparrow$) indicates higher score is better while ($\downarrow$) indicates that lower is better. ✗ indicates that value diverged towards $-\infty$. Switch events for decomposed models are defined as times where the active set of DOs change from the previous time step.

| Model | NASCAR | | | Lorenz | | |
|---|---|---|---|---|---|---|
| | Dynamics MSE ($\downarrow$) ($\times10^{-3}$) | Switch MSE ($\downarrow$) ($\times10^{-3}$) | 100-step $R^2$ ($\uparrow$) | Dynamics MSE ($\downarrow$) | Switch MSE ($\downarrow$) | 100-step $R^2$ ($\uparrow$) |
| SLDS | 0.0995 | 12.89 | 0.184 | 0.431 | 0.0204 | -3.47 |
| rSLDS | 0.1065 | 13.17 | 0.238 | 0.304 | 0.0208 | -11.54 |
| dLDS | 123.19 | 13.28 | ✗ | 1.123 | 0.1529 | ✗ |
| p-dLDS (ours) | **0.033** | **7.34** | **0.450** | **0.141** | **0.0137** | **0.418** |

## 4.1 Synthetic Dynamical Systems

**NASCAR with Random Speeds.** We evaluate our inference procedure on the NASCAR dataset [26, 32, 23]. Since this system is easily mean-centered, we can isolate the effect of our proposed inference procedure as offset terms are not necessary. To make this dataset more realistic, we introduce speed variability into the ground truth dynamics as opposed to having a perfect constant speed at all time. Specifically, whenever a trajectory enters a new segment of the track, the system experiences a random change in speed. We trained all models on 30 trials, each consisting of 1000 time steps (Fig. 2A) with randomly sampled initial points, and a randomly constructed 10-dimensional linear observation matrix (see Appendix C.1 for full details). Performance is evaluated on 30 held-out trials where the ground truth switching states are defined by the different segments of the track. In all models, we set the $M = 10$, $N = 2$ and $K = 4$ for DOs or switching states. In our experiments, we define the "discrete states" for decomposed models as the DO state with the largest coefficient magnitude.

Figure 2B shows that changes in the system speed mask the true transition behavior between segments of the track in rSLDS. Moreover, dLDS identifies coherent segments, but inappropriately learns a different switching pattern for outer and inner edges of the track. In contrast, p-dLDS identifies a switching pattern most consistent with the true track segments despite the presence of noise and randomness in the system's speed. We note that while there are four true segments, decomposed models form a more parsimonious representation by identifying similar behaviors in different track segments such as in both edges and curves.

Table 1 summarizes our quantitative evaluations on three metrics: 1) the mean squared error (MSE) between the learned and ground truth latent dynamics, 2) the MSE between the inferred and true switch rate to determine agreement of the discrete switching behavior, and 3) the 100-step inference $R^2$ to demonstrate that the learned system generalizes beyond a single step on held-out data. (See Appendix D for mathematical definitions). We see that p-dLDS broadly outperforms existing methods in all metrics and significantly improves inference for decomposed models.

**Lorenz System with Random Ramping.** Next, we consider the Lorenz system, a chaotic nonlinear system with multiple fixed points, exploring the effect of the offset term. The system is described by the differential equation, $\dot{\boldsymbol{x}} = [\sigma(x_2 - x_1), x_1(\rho - x_3) - x_2, x_1 x_2 - \beta x_3]^\top$ where the parameters $\rho = 28$, $\beta = 8/3$, and $\sigma = 10$ define a chaotic attractor with two opposing lobes (Fig. 2D). We introduce continuous fluctuations in the underlying dynamics by randomly ramping the system's speed throughout each trajectory. This is accomplished by adjusting the evaluation time intervals given to an ODE solver. Similar to before, we randomly construct a linear observation function with $M = 10$ and train models on 30 randomly constructed trials with 1000 time points each (see Appendix C.2). Furthermore, we define the ground truth switch events as the time points when the signal transitions between the two lobes in addition to the moments when a ramping period concludes. All models are trained with a latent space of $N = 3$ and $K = 4$ states or DOs.

In figure 2D, we see that rSLDS does not distinguish between the different speeds along the outer and inner sections of the attractor. Instead, the discrete states obscure the continuum of speeds by incorrectly grouping all activity in each lobe into a single regime. Furthermore, we observe that dLDS is limited without an offset term, unable to accurately represent multiple fixed points. Instead of aligning with the two attractor lobes, transitions in the dominant coefficients occur radially relative to the origin and fail to reconstruct the two orbiting fixed points. Conversely, p-dLDS's offset term enables learning a system where coefficients better match the true geometry. This representation correctly recovers differences between the outer and inner sections of the attractor while also accurately reconstructing the two orbiting fixed points. Moreover, this leads to improved estimation of latent dynamics, a switching rate that agrees with the true system, and improved multistep inference performance as shown in Table 1.

## 4.2   Simulated Motor Cortex Data in a Reaching Task

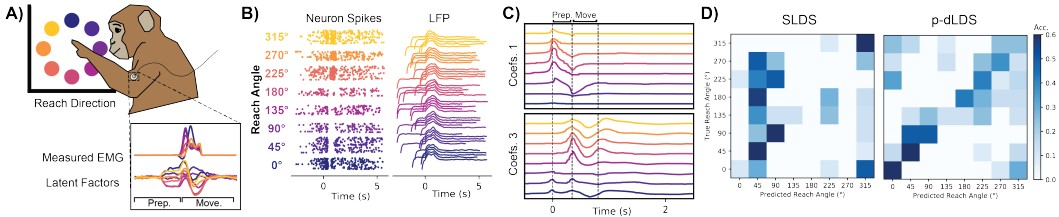

Figure 3: **p-dLDS efficiently captures changes in dynamics. (A)** Latent factors are computed from empirical EMG of the reaching experiment in [8]. Dynamics are characterized by a preparatory and movement phase. **(B)** Synthetic spikes and LFPs are generated using the `wslfp` package [17, 16] **(C)** The trial-averaged coefficients for p-dLDS smoothly vary with reaching angle. DO 1 captures preparatory dynamics while DO 3 captures movement dynamics. **(D)** Confusion matrix for linear classification of reach directions. p-dLDS predictions closely align to true diagonal.

We now turn to an empirically-derived synthetic experiment related to brain-computer interfaces, where the dynamics and observation functions are nonlinear and derived from analysis of neural data. Our focus is on the reaching task, a neuroscience experiment designed to study motor control in non-human primates [19, 22]. In this experiment, the subjects are trained to reach towards visually cued targets, while neural activity is recorded from motor-related areas such as electromyography (EMG) data from arm muscles. Each trial consists of two distinct phases: preparation and movement. In the preparation phase, the subject plans its movement while keeping their arm still. In the movement phase, the subject physically reaches towards the target. The goal of this experiment is to decode reach intention from neural data. We construct a dataset by first simulating a spiking neural network with known latent factors [8] trained to reproduce empirical EMG signals from the center-out reach

Table 2: Inference performance for the reaching experiment (see Figure 3) on a held-out test set. Top-1 and Top-3 accuracies are obtained by predicting reach directions from latent variable features using linear classifiers. State and Dynamics MSE are computed with respect to true latent variables.

| Model | Top-1 Acc. ($\uparrow$) | Top-3 Acc. ($\uparrow$) | State MSE ($\downarrow$) ($\times 10^{-1}$) | Dynamics MSE ($\downarrow$) ($\times 10^{-2}$) |
|---|---|---|---|---|
| SLDS | 38.46 | 57.69 | 0.5289 | 0.3942 |
| rSLDS | 12.82 | 32.05 | 0.5503 | 292.41 |
| dLDS | 10.25 | 39.74 | 0.6742 | 35.680 |
| pdLDS (ours) | **42.31** | **70.51** | **0.4061** | **0.0567** |

task in [22]. Spikes are then converted to 50-channel local field potentials (LFP) recordings via a weighted, delayed sum of synaptic currents (see Figure 3A and B) [29, 17, 16]. Our dataset contains 150 6-second trials sampled at 250 hz, where each trial represents one out of eight reach directions visually cued at a random start time. PCA identifies that three components captures 98% of the variance. Thus, we set $M = 50$, $N = 3$, and $K = 4$ DOs or discrete states.

Figure 3C shows the trial-averaged DO coefficients from p-dLDS, which change smoothly and cyclically according to the true reach angle. Additionally, the DOs appear to differentiate between the two distinct dynamical regimes, where the activity of $f_1$ and $f_3$ localize to the preparatory and movement phase respectively. Quantitatively, we compute the linear classification accuracy of the reach angles using the average state activity over time as features (see Appendix D.5). Figure 3D shows that classifiers built from SLDS features fail to capture the full continuum of reach angles. This limitation occurs because the discrete switching states are unable to efficiently capture the diversity of activity present in the LFPs, which arise from randomness inherent in the spike sampling process. Consequently, the inferred features from SLDS generalize poorly to held-out data. In contrast, the p-dLDS classifier predictions recover the full spectrum of reach angles since features are naturally continuous and the inferred coefficients can adjust the learned DOs to accurately capture the activity in the held out data. Table 2 shows that p-dLDS outperforms all other models in state and dynamics reconstruction as well as the top-1 and top-3 reach classification accuracy.

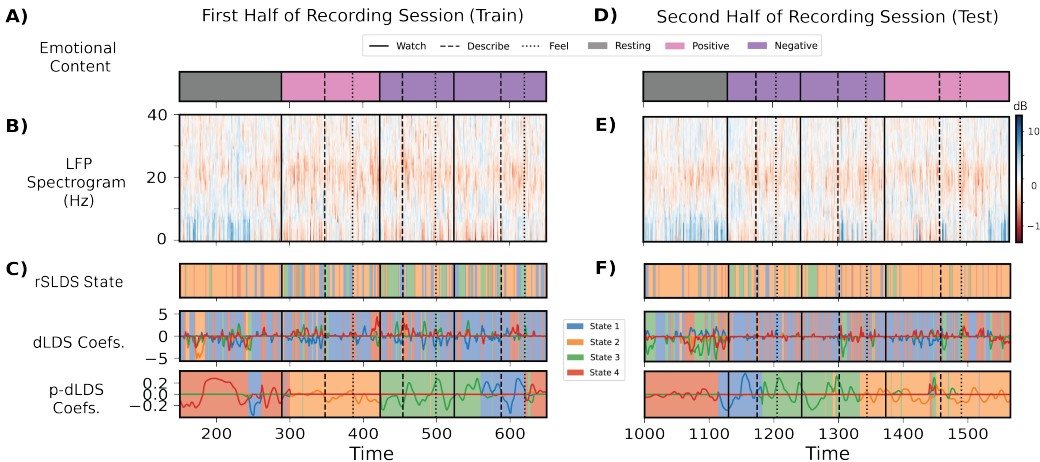

Figure 4: **Learned system discovers coherent structure in clinical data.** **(A, D)** LFP data was collected on patients watching videos with different emotional content. **(B, E)** LFP spectrograms are 40-dimensional signals where each channel represents a particular frequency. **(C, F)** Inferred states and coefficients shows that rSLDS and dLDS exhibit unpredictable switching behavior. In contrast, p-dLDS captures smooth coefficients and identifies DOs that align with the trial's emotional content. The learned patterns broadly generalize to the held-out data.

### 4.3 Clinical Neurophysiology Data

We demonstrate p-dLDS on LFP recordings from the subcallosal cingulate cortex (SCC) in patients with treatment-resistant depression (ClinicalTrials.gov identifier NCT01984710). Subjects are asked to watch videos with different emotional content (positive, negative, and neutral), describe the videos, and then discuss how the video made them feel. SCC dynamics have been previously shown to provide a quantitative signal for the presence of emotional content [14] and depression recovery [1]. Thus, we hypothesize that the underlying dynamics may provide information about emotional changes throughout the experiment. We apply p-dLDS to a single patient's LFP spectrogram data (Fig. 4B, E) within the 0-40 Hz frequency range ($M = 40$). PCA indicates that the first 7 components explain 90% of the variance. Therefore, we train a model with $N = 7$ latent dimension and $K = 4$ DOs.

In Figure 4, rSLDS and dLDS produces a high degree of state oscillations making it difficult to identify time intervals with consistent emotional content. In contrast, p-dLDS infers coherent structure that corresponds to changes in emotional content in the trial. For example, $f_4$ (red) coincides with resting, $f_2$ (orange) with positive videos, and $f_1$ and $f_3$ (blue and green) to negative videos (Fig. 4C). Importantly, this structure persists even on held out data from the second half of the session (Fig. 4 F). We note this preliminary analysis on a single subject isn't intended to make a claim about specific neurophysiological responses to emotional content in this brain region, but generally highlights that p-dLDS identifies meaningful dynamical modes where previous models are unable to.

## 5 Conclusion

In this work, we present a probabilistic decomposed linear dynamical systems model that can be used to discover meaningful representations in neural signals. By marginalizing over uncertainty in latent variable estimates and incorporating an offset into the dynamics, we enhance robustness and improve a variety of performance metrics. Some areas of future work includes exploiting structure in the offsets to automatically identify window size and extending the probabilistic model to include more complicated emissions distributions, such as the Poisson likelihood commonly used to model neural spiking data [28].

## Acknowledgements and Disclosure of Funding Sources

Y.C. and C.R. were funded by the James S. McDonnell Foundation (grant number 22002039), with Y.C. being further funded by National Institutes of Health (grant number 2T32EB025816), and C.R. being further funded by the Julian T. Hightower Chair. Y.C. and K.J. were part of the Georgia Tech/Emory NIH/NIBIB Training Program in Computational Neural Engineering (T32EB025816). N.M. was funded by The Kavli Foundation NeuroData Discovery award. A.S.C. were partially supported by the NSF CAREER Award (2340338) and a Johns Hopkins Bridge Grant. S.A. is supported by the National Center for Advancing Translational Sciences of the National Institutes of Health (Award Number UL1TR002378 and KL2TR002381). The content is solely the responsibility of the authors and does not necessarily represent the official views of the National Institutes of Health.

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

# A   Appendix / supplemental material

## A.1   p-dLDS Algorithm

Algorithm 1 describes the proposed inference algorithm. In our experiments, we set $n = 1$ and $\eta = 10^{-4}$ and observe that the model converges. Below we use the notation hat notation for latent variable estimates or samples and the variable itself to represent the parameters of the variational distributions.

---

**Algorithm 1** Variational EM for Probabilistic dLDS

---

**Require:** $M$ observation dimension, $N$ latent state dimension, $K$ number of dynamic operators, $S$ moving average window size, $\xi$ SBL-DF trade-off parameter, $n$ number of samples to estimate expectations, $\eta$ sparsity threshold, $\theta$ model parameters.

  **// Initialize parameters**
$\boldsymbol{c}_t \leftarrow \mathbf{0}$
$D_{i,j} \sim \mathcal{N}(0, \sigma^2)$
$f_{k,i,j} \sim \mathcal{N}(0, \sigma^2)$
$\widehat{\boldsymbol{x}}_t \leftarrow \boldsymbol{D}^+ \boldsymbol{y}_t$             ▷ Initialize latent state with PCA

  **while** ELBO has not converged **do**
    **// Update Latent State Posterior**
    $\widehat{\boldsymbol{b}}_{1:T} \leftarrow \text{MovingAverage}_S(\widehat{\boldsymbol{x}}_{1:T})$
    $\widehat{\boldsymbol{c}}_t \sim q(\boldsymbol{c}_t)$
    $\widehat{\boldsymbol{F}}_t \leftarrow \sum_{k=1}^{K} \boldsymbol{f}_k \widehat{c}_{k,t}$
    $\boldsymbol{l}_{1:T}, \boldsymbol{\Sigma}_x \leftarrow \text{KalmanSmoother}(\boldsymbol{y}_{1:T}, \widehat{\boldsymbol{b}}_{1:T}, \widehat{\boldsymbol{F}}_{1:T}, \theta)$

    **// Update Coefficient Posterior**
    Initialize $q(\boldsymbol{c})$ and $q(\boldsymbol{\gamma})$ jointly with SBL-DF.
    Update $q(c_{t,k})$ with SGD over equation (10) for densities where $|c_{t,k}| > \eta$.
    Update $q(\gamma_t) \leftarrow \mathcal{IG}(\xi + \frac{n}{2}, \xi c_{t-1,k}^2 + \frac{\sum_{i=1}^{n}(\tilde{c}_{t,k,i} - c_{t,k})^2}{2})$

    **// Update Parameters**
    Update $\theta$ with SGD over equation (11).
  **end while**

---

# B   Latent Variable Inference

## B.1   Lemma 1 Derivation

**Lemma 1.** *Let the transition between any two state vectors $\boldsymbol{x}_t, \boldsymbol{x}_{t+1} \in \mathbb{R}^N$ be defined by the linear dynamics matrix $\boldsymbol{F}_t \in \mathbb{R}^{N \times N}$ and the dynamics offset $\boldsymbol{b}_t \in \mathbb{R}^N$. For any $\lambda > 0$, the objective,*

$$\arg\min_{\boldsymbol{F}_t, \boldsymbol{b}_t} \|\boldsymbol{x}_{t+1} - \boldsymbol{x}_t - \boldsymbol{F}_t \boldsymbol{x}_t - \boldsymbol{b}_t\|_2^2 + \lambda \|\boldsymbol{F}_t\|_2^2,$$

*is minimized when $\boldsymbol{F}_t = \mathbf{0}$ and $\boldsymbol{b}_t = \boldsymbol{x}_{t+1} - \boldsymbol{x}_t$.*

*Proof.* Let $\boldsymbol{r}_t = \boldsymbol{x}_{t+1} - \boldsymbol{x}_t$. We can rewrite the reconstruction objective in the following form,

$$\arg\min_{\boldsymbol{F}_t, \boldsymbol{b}_t} \|\boldsymbol{r}_t - \boldsymbol{F}_t \boldsymbol{x}_t - \boldsymbol{b}_t\|_2^2 + \lambda \|\boldsymbol{F}_t\|_2^2.$$

This objective is identical to the standard ridge regression with an unpenalized intercept term [13]. The solution is obtained by first centering the data, and then solving for the parameters using the solution for the standard Tikhonov regression. Below, we define the centered data as $\tilde{\boldsymbol{x}}_t$ and $\tilde{\boldsymbol{r}}_t$ for inputs and outputs respectively. Finally, we can use these values to obtain the following estimates of the parameters,

$$\widehat{\boldsymbol{b}}_t = \boldsymbol{\mu}_t \qquad \widehat{\boldsymbol{F}}_t = (\tilde{\boldsymbol{x}}_t^\top \tilde{\boldsymbol{x}}_t + \lambda \boldsymbol{I})^{-1} \tilde{\boldsymbol{x}}_t^\top \tilde{\boldsymbol{r}}_t$$

However, when there is only a single datapoint, we get that $\tilde{x}_t = 0$, which results in $\widehat{\boldsymbol{F}} = 0$. $\qquad \square$

This result arises from having only a single observation for any dynamic transition, which leads to a singular design matrix. Although we can improve our estimate of $\boldsymbol{F}_t$ by collecting more samples along a given trajectory, this is impractical when dealing with naturalistic time-series. For instance, it may be infeasible to collect more data from the exact same initial condition in a naturalistic environment due to noise in the experimental setup. In chaotic systems, minor deviations can lead to drastically different outcomes over long time horizons. Even if it were possible to precisely control for the initial condition of the signal, the presence of dynamical noise can cause initially aligned time series to quickly drift out of alignment. Consequently, it is not uncommon to observe a single transition between any two time points, as it is not guaranteed that events across multiple trials will be well-aligned.

### B.2 Lemma 2 Derivation

**Lemma 2.** *Let $\boldsymbol{l}, \boldsymbol{b} \in \mathbb{R}^N$ be independent random variables such that $\boldsymbol{l} \sim p(\boldsymbol{l})$ and $\boldsymbol{b} \sim p(\boldsymbol{b})$. Their sum $\boldsymbol{x} = \boldsymbol{l} + \boldsymbol{b}$ is distributed according to $\mathcal{N}(\boldsymbol{\mu}_l + \boldsymbol{\mu}_b, \boldsymbol{\Sigma}_l + \boldsymbol{\Sigma}_b)$ when 1) $p(\boldsymbol{b}) = \mathcal{N}(\boldsymbol{\mu}_b, \boldsymbol{\Sigma}_b)$ and $p(\boldsymbol{l}) = \mathcal{N}(\boldsymbol{\mu}_l, \boldsymbol{\Sigma}_l)$ and when 2) $p(\boldsymbol{b}) = \delta(\boldsymbol{b} - \boldsymbol{\mu}_b)$ and $p(\boldsymbol{l}) = \mathcal{N}(\boldsymbol{\mu}_l, \boldsymbol{\Sigma}_l + \boldsymbol{\Sigma}_b)$.*

*Proof.* **Case 1.** Let $\boldsymbol{l} \sim \mathcal{N}(\boldsymbol{\mu}_l, \boldsymbol{\Sigma}_l)$ and $\boldsymbol{b} \sim \mathcal{N}(\boldsymbol{\mu}_b, \boldsymbol{\Sigma}_b)$. The sum of normal random variables follows a distribution that results from convolving their individual distributions,

$$\begin{aligned} q(\boldsymbol{x}) &= q(\boldsymbol{l} + \boldsymbol{b}) \\ &= q(\boldsymbol{l}) * q(\boldsymbol{b}) \\ &= \mathcal{N}(\boldsymbol{\mu}_l, \boldsymbol{\Sigma}_l) * \mathcal{N}(\boldsymbol{\mu}_l, \boldsymbol{\Sigma}_b) \\ &= \mathcal{N}(\boldsymbol{\mu}_l + \boldsymbol{\mu}_b, \boldsymbol{\Sigma}_l + \boldsymbol{\Sigma}_b) \end{aligned}$$

This is a standard result from probability theory.

**Case 2.** Now let $\boldsymbol{l} \sim \mathcal{N}(\boldsymbol{\mu}_l, \boldsymbol{\Sigma}_l + \boldsymbol{\Sigma}_b)$ and $\boldsymbol{b} \sim \delta(\boldsymbol{b} - \boldsymbol{\mu}_b)$. Similarly, the distribution of the sum of these variables is distributed according to their convolution,

$$\begin{aligned} q(\boldsymbol{x}) &= q(\boldsymbol{l}) * q(\boldsymbol{b}) \\ &= \mathcal{N}(\boldsymbol{\mu}_l, \boldsymbol{\Sigma}_l + \boldsymbol{\Sigma}_b) * \delta(\boldsymbol{b} - \boldsymbol{\mu}_b) \\ &= \int_{-\infty}^{\infty} \mathcal{N}(\boldsymbol{x} - \boldsymbol{\tau}; \boldsymbol{\mu}_l, \boldsymbol{\Sigma}_l + \boldsymbol{\Sigma}_b) \delta(\boldsymbol{\tau} - \boldsymbol{\mu}_b) d\boldsymbol{\tau} \\ &= \mathcal{N}(\boldsymbol{x} + \boldsymbol{\mu}_b; \boldsymbol{\mu}_l, \boldsymbol{\Sigma}_l + \boldsymbol{\Sigma}_b) \\ &= \mathcal{N}(\boldsymbol{x}; \boldsymbol{\mu}_l + \boldsymbol{\mu}_b, \boldsymbol{\Sigma}_l + \boldsymbol{\Sigma}_b), \end{aligned}$$

where the fourth line is the result of the sifting property of delta distributions. Since the final distribution in Case 1 and Case 2 are identical, we complete the proof. $\qquad \square$

### B.3 Optimal $q(x)$ Update

The optimal coordinate ascent variational update is given by the following equation,

$$\begin{aligned} \log q^*(\boldsymbol{x}) &\propto \mathbb{E}_{q(\boldsymbol{c}, \boldsymbol{\gamma})}[\log p(\boldsymbol{x}, \boldsymbol{c}, \boldsymbol{y}, \boldsymbol{\gamma}|\theta)] \\ &= \mathbb{E}_{q(\boldsymbol{c}, \boldsymbol{\gamma})}\Big[\log p(\boldsymbol{l}_1|\theta) + \sum_{t=2}^{T} \log p(\boldsymbol{l}_t|\boldsymbol{l}_{t-1}, \boldsymbol{c}_t, \theta) + \sum_{t=1}^{T} \log p(\boldsymbol{y}_t|\boldsymbol{l}_t + \boldsymbol{b}_t, \theta)\Big] + C. \end{aligned} \tag{12}$$

Conditioned on estimates of $\boldsymbol{b}_{1:T}$ and samples of $\boldsymbol{c}_{1:T}$, the factor graph of equation (12) corresponds exactly to a time-varing Linear Gaussian State Space Model. Thus we can leverage the efficient inference algorithms such as the Kalman filter and RTS smoother when computing the marginals of the variational distribution of $\boldsymbol{l}_{1:T}$.

## C    Generating Synthetic Examples

### C.1    Noisy NASCAR

NASCAR data is generated by partitioning the two-dimensional state space into four regions according to the rules,

$$Z(\boldsymbol{x}) = \begin{cases} 1, & x_1 > 1 \\ 2, & x_1 < -1 \\ 3, & -1 \leq x_1 \leq 1, x_2 \geq 0 \\ 4, & -1 \leq x_1 \leq 1, x_2 < 0, \end{cases}$$

where $Z(\boldsymbol{x})$ is the ground truth switching state function that depends on the particular location $\boldsymbol{x}$. The ground truth dynamics matrices are defined as,

$$\boldsymbol{A}(\boldsymbol{x}) = \begin{cases} \begin{bmatrix} 0 & 0.1 \\ -0.1 & 0 \end{bmatrix}, & \text{when } Z(\boldsymbol{x}) = 1 \text{ or } 2 \\ \begin{bmatrix} 0 & 0 \\ 0 & 0 \end{bmatrix}, & \text{when } Z(\boldsymbol{x}) = 3 \text{ or } 4, \end{cases}$$

and ground truth offsets are defined as,

$$\boldsymbol{b}(\boldsymbol{x}) = \begin{cases} \begin{bmatrix} 0 & 0.005 \end{bmatrix}^\top, & \text{when } Z(\boldsymbol{x}) = 1 \\ \begin{bmatrix} 0 & -0.005 \end{bmatrix}^\top, & \text{when } Z(\boldsymbol{x}) = 2 \\ \begin{bmatrix} 0.1 & 0 \end{bmatrix}^\top, & \text{when } Z(\boldsymbol{x}) = 3 \\ \begin{bmatrix} -0.1 & 0 \end{bmatrix}^\top, & \text{when } Z(\boldsymbol{x}) = 4. \end{cases}$$

Given the current location in state space $\boldsymbol{x}_t$, we can transition to the next point using the continuous time dynamics equation

$$\boldsymbol{x}_t = \text{expm}(\tau \boldsymbol{A}_{Z(\boldsymbol{x}_t)})\boldsymbol{x}_{t-1} + \tau \boldsymbol{b}_{Z(\boldsymbol{x}_t)} + \boldsymbol{\nu}_t,$$

where each entry of the process noise is sampled from $\nu_{t,i} \sim \mathcal{N}(0, 10^{-4})$. To modulate the speed of the system, we uniformly sample a speed constant $\tau \in [0.1, 1]$, which is applied throughout each segment of the track. We use the continuous time formulation over the discrete-time formulation to ensure that changes to the speed do not distort the shape of the original system's state space. To generate noisy observations, we construct a linear emissions matrix with random variables such that each entry is given by $D_{i,j} \sim \mathcal{N}(0, 1)$.

### C.2    Ramping Lorenz

In order to modulate the speed of the Lorenz system, we adjust the evaluation time points of an ODE integrator, specifically Runge-Kutta of the order 5(4) (RK54) as implemented in scipy's `solve_ivp` [9]. Ramping activity is generated randomly with the following procedure,

1. Uniformly sample an evaluation interval length $\tau \in [0.25, 1.5]$.

2. Construct a vector $\widetilde{T}$ that consists $n$ evenly spaced numbers over the interval $[0, \tau]$. In our experiments, we set $n$ to be 100.

3. Perform the transformation $\exp(\widetilde{T}) - 1$ to obtain a vector of ramped evaluation times.

4. Plug in the transformed evaluation times into the RK45 Solver to obtain latent trajectories.

Similar to the NASCAR experiment, we generate noisy observation from a randomly constructed linear emissions matrix such that each entry is given by $D_{i,j} \sim \mathcal{N}(0, 1)$.

## C.3 Simulated Monkey Reaching Task

Our dataset is constructed from publicly available data and code from the center-out reach task in [22, 8]. We obtain latent factors from spiking networks that are trained to reproduce empirically measured EMG signals, given a 3-dimensional input that specifies the go input and the reach angle. In our experiments, these factors are considered ground truth. Our trained factor-based spiking network then generates spiking activity for 1200 neurons. Synaptic currents are used as inputs into the Weighted Sum of synaptic currents LFP proxy method (WSLFP) [29], as implemented in the wslfp Python package [17, 16]. As WSLFP is a function of the relative location of neurons and electrodes, we place neurons randomly within a 5 mm by 10 mm by 1 mm region and electrodes in a grid centered in this region. The result is a multi-channel LFP dataset with nonlinear dynamics and measurements characteristic of systems neuroscience.

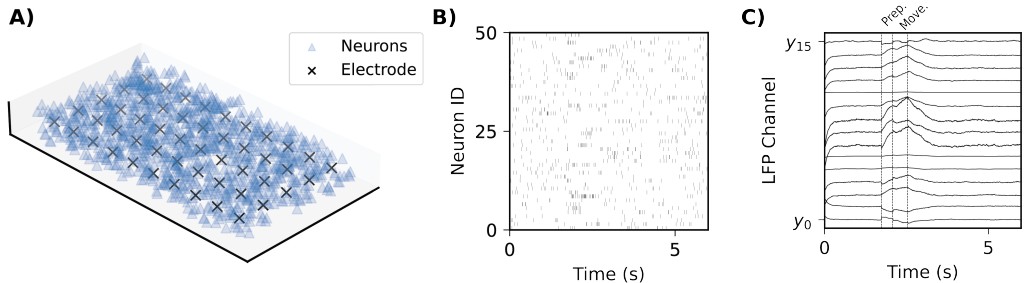

Figure 5: **Empirically-Derived Reach Experiment. (A)** 1,200 neurons are randomly placed into a 5 mm by 10 mm by 1 mm region. Electrodes are placed in a grid centered in this region **(B)** Spiking activity for a subset of neurons in an example trial produced from a factor-based spiking network. **(C)** First 15 channels in a simulated multi-channel LFP recording. Preparatory and Movement phases are marked by the dotted lines.

# D Evaluation Metrics

## D.1 Multi-step Inference

The multi-step inference performance is computed with the following R-squared metric,

$$R_k^2 = 1 - \frac{\sum_{t=0}^{T-k} \|\boldsymbol{y}_{t+k} - \widehat{\boldsymbol{y}}_{t+k}\|_2^2}{\sum_{t=0}^{T-k} \|\boldsymbol{y}_{t+k} - \bar{\boldsymbol{y}}\|_2^2}, \tag{13}$$

where $k$ is the number of steps from the initial condition, $\bar{\boldsymbol{y}}$ is the mean estimator for each trajectory and $\widehat{\boldsymbol{y}}_{t+k}$ is the model prediction after applying the inferred dynamics for $k$ steps. When testing, model parameters such as the dynamics and observation matrices are frozen, while specific latent variables are estimated based on the held-out data. In Table 1, we show results for $k = 100$.

## D.2 Inferred Dynamics Error

We measure the accuracy of the latent dynamics with the mean squared error (MSE) of the inferred speed, defined as,

$$\text{MSE}_{\text{speed}} = \frac{1}{T-1} \sum_{t=1}^{T-1} \|\dot{\boldsymbol{x}}_t - \boldsymbol{U}\widehat{\boldsymbol{x}}_t\|_2^2, \tag{14}$$

where the true speed $\dot{\boldsymbol{x}}_t = \boldsymbol{x}_{t+1} - \boldsymbol{x}_t$ is computed from the denoised ground truth latent state, and the predicted speed $\widehat{\dot{\boldsymbol{x}}}_t = \widehat{\boldsymbol{x}}_{t+1} - \widehat{\boldsymbol{x}}_t$ is computed using the model's 1-step prediction. Since latent trajectories are only identifiable up to a linear transformation, we align the inferred trajectories with the true trajectories using a least squares fit before computing this score. More specifically, we

find the optimal linear transformation $\boldsymbol{U} \in \mathbb{R}^{N \times N}$ between the estimated and true states across all trajectories by solving,

$$\widehat{\boldsymbol{U}} = \arg\min_{\boldsymbol{U}} \frac{1}{T} \sum_{t=1}^{T} \|\boldsymbol{x}_t - \boldsymbol{U}\widehat{\boldsymbol{x}}_t\|. \tag{15}$$

### D.3 Inferred Latent State Space Error

Similarly, we measure the accuracy of the latent state space by computing the MSE after a linear alignment between trajectories from the inferred and true state space. We use this metric only for the reaching example, since the true observation function is a complex nonlinear function,

$$\text{MSE}_{\text{state}} = \frac{1}{T} \sum_{t=1}^{T} \|\boldsymbol{x_t} - \boldsymbol{U}\widehat{\boldsymbol{x}}_t\|_2^2. \tag{16}$$

The linear alignment $\boldsymbol{U} \in \mathbb{R}^{N \times N}$ between the estimated and true states across all trajectories is computed by solving the least squares problem in equation (15).

### D.4 Inferred switching rate error

Evaluating the accuracy of the switching behavior is a more difficult task. In fact, developing a procedure that matches predicted switch times with true switch times can lead to a complicated optimal transport procedure. To simplify the evaluation of switching times, we marginalize over time, and compare only the MSE of the switch rate defined as,

$$\text{MSE}_{\text{switch}} = \frac{1}{m} \sum_{i=1}^{m} \|r_i - \widehat{r}_i\|_2^2, \tag{17}$$

where $m$ is the number of trials, $r_i$ is the true switch rate for the $i$th trajectory, and $\widehat{r}_i = \frac{1}{T} \sum_{t=1}^{T} \mathbf{1}\{z_t \neq z_{t-1}\}$ is the predicted switch rate. Intuitively, $\widehat{r}_i$ is the number of times that the state or dominant DO changes between consecutive time points normalized by the length of the interval $T$. In switching models, switch events are defined as a time point where the current inferred dynamical state differs from the state in the previous time step. Similarly in decomposed models, switch events are defined as time points where the active set of DOs change from the previous time step.

In the NASCAR example, $r_i$ is defined with the number of transitions between ground truth segments. In the Lorenz example, $r_i$ is defined by the number of times that the trajectory switches between the two lobes in addition to the number of ramping periods.

### D.5 Reaching Classification Accuracy

We quantitatively evaluate the reaching experiment with a classification task. Here, we want to determine whether the learned systems can be used to distinguish between different reach directions. Recall that switched models infer a switching variable for each time point where $z_t \in \{1, \ldots, K\}$ while decomposed models infer a coefficient vector $\boldsymbol{c}_t \in \mathbb{R}^K$. Rather than viewing $z_t$ as an index, we can equivalently view it as a one-hot encoded vector $z_t \in \{0, 1\}^K$ which describes whether a particular switching state is active at any given time. This matches the dimensionality of the variables in both switched and decomposed systems.

For simplicity, we focus on linear logistic regression classifiers in our experiment. If we let the inputs be $z_t$ and $c_t$ directly, then our classifiers quickly overfits since there are many more input features than trials. Specifically, the number of features scales linearly with the number of time points and systems $\mathcal{O}(TK)$. Instead, we marginalize over time and compute features from the estimated latent variables by averaging state activity over time. In switched models, this is the average one-hot encoding value over time. Similarly, this is the average coefficient value in decomposed models. However, for each dynamical state, we compute separate features for positive and negative coefficient values to prevent interference between them. In this setup, the input (feature) dimensionality scales according to $\mathcal{O}(K)$ while the output dimensionality of the linear classifiers are the reaching directions. For all

classifiers, we perform a grid search over the values $\{10^i\}_{i=-4}^{4}$ to identify an appropriate amount of L2 regularization. Top-k accuracies are a standard metric in machine learning [21, 3] and computed using the estimated class probabilities from the logistic regression classifier.

# E  Additional Results

## E.1  Synthetic Dynamical Systems

Figure 6A demonstrates that our inference procedure converges to a local optimium while Figure 6B shows a full sweep of the multi-step inference metric. Tables 1 in the main paper reports the final value. For completeness, we include Tables 3 and 4 which reports the means across 5 seeds of each model, and includes the standard deviations in parenthesis.

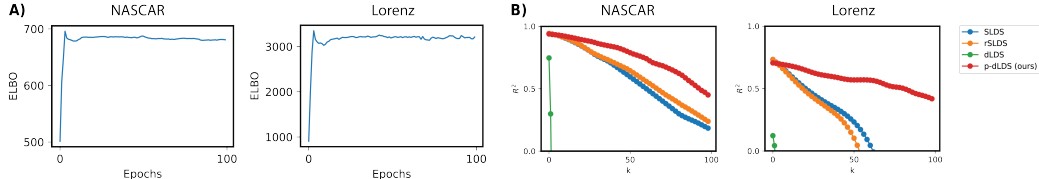

Figure 6: **(A)** ELBO converges in both synthetic dynamical systems. **(B)** Multi-step inference where $k$ represents the number of steps. Tables 3 and 4 report the final values.

Table 3: Metrics for NASCAR. Bold means best performance. ($\uparrow$) indicates higher score is better while ($\downarrow$) indicates that lower is better. ✗ indicates that value diverged towards $-\infty$. All MSE values are $\times 10^{-3}$ while $R^2$ values are not scaled. We report means across 5 seeds and include standard deviation in parenthesis.

| Model | Speed MSE ($\downarrow$) | Switch MSE ($\downarrow$) | 100-step $R^2$ ($\uparrow$) |
|---|---|---|---|
| SLDS | 0.0995 (0.021) | 12.89 (1.30) | 0.184 (0.024) |
| rSLDS | 0.1065 (0.024) | 13.17 (2.84) | 0.238 (0.022) |
| dLDS | 123.19 (23.13) | 13.28 (5.31) | ✗ |
| p-dLDS (ours) | **0.033** (0.009) | **7.34** (3.40) | **0.450** (0.027) |

Table 4: Metrics for Lorenz. Bold means best performance. ($\uparrow$) indicates higher score is better while ($\downarrow$) indicates that lower is better. ✗ indicates that value diverged towards $-\infty$. We report means across 5 seeds and include standard deviation in parenthesis.

| Model | Speed MSE ($\downarrow$) | Switch MSE ($\downarrow$) | 100-step $R^2$ ($\uparrow$) |
|---|---|---|---|
| SLDS | 0.431 (0.233) | 0.0204 (0.007) | -3.47 (1.052) |
| rSLDS | 0.304 (0.040) | 0.0208 (0.004) | -11.54 (1.353) |
| dLDS | 1.123 (0.089) | 0.1529 (0.070) | ✗ |
| p-dLDS (ours) | **0.141** (0.015) | **0.0137** (0.014) | **0.418** (0.079) |

## E.2  Reaching Task

For each model, we visualize the trial-averaged dynamic regime activity of each reach direction (Fig. 7). In SLDS, this is visualized by considering the discrete states as a one hot vector over time. When a dynamic regime is active, that state will have a value of 1 while the unactive states will have a value of 0. Thus the trial averaged value of each state must have a value in the interval $[0, 1]$. In dLDS, we plot the inferred coefficient value without any modification.

Although SLDS correctly identifies preparatory and movement phases using states 4 and 3 respectively, it fails to differentiate dynamics occurring outside of these expected phases, incorrectly grouping

unrelated regions together. Furthermore, the discrete formulation produces very similar activity patterns across all reach angles, obscuring any differences that are present. In dLDS, we observe that the features change smoothly and cyclically with the reach angle. However, the dynamic operator activity do not localize to the preparatory and movement phases due to a limited inference procedure.

Table 5: Inference performance for the reaching experiment (see Figure 3) on a held-out test set. Top-1 and Top-3 accuracies are obtained by predicting reach directions from latent variable features using linear classifiers. State and Dynamics MSE are computed with respect to true latent variables. We report standard deviations in parenthesis across 5 seeds.

| Model | Top-1 Acc. | Top-3 Acc. | State MSE ($\times 10^{-1}$) | Dynamics MSE ($\times 10^{-2}$) |
|---|---|---|---|---|
| SLDS | 38.46 (2.84) | 57.69 (7.53) | 0.5289 (0.13) | 0.3942 (0.23) |
| rSLDS | 12.82 (3.05) | 32.05 (8.31) | 0.5503 (0.23) | 292.41 (13.96) |
| dLDS | 10.25 (5.97) | 39.74 (10.29) | 0.6742 (0.52) | 35.680 (5.76) |
| pdLDS (ours) | **42.31** (3.50) | **70.51** (6.45) | **0.4061** (0.38) | **0.0567** (0.04) |

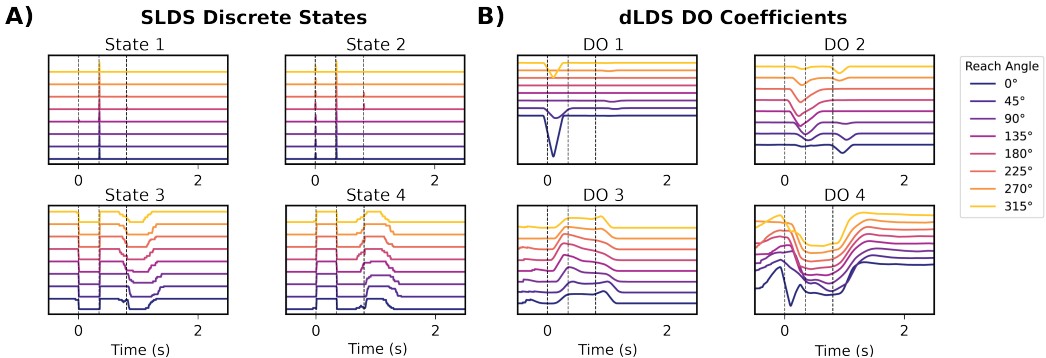

Figure 7: Trial-averaged activity for **(A)** SLDS discrete states and **(B)** dLDS DO coefficients for each reach angle. The preparatory and movement phases occur between the dashed lines similar to Figure 3. Time 0 represents the onset of the stimulus.

# F  Experimental Setup

## F.1  Hyperparameter Settings

For switching models, we rely on the `ssm` package which allows for efficient Bayesian inference for a variety of state space models [25]. We set the variational posterior to `structured_meanfield`, and the fitting procedure to `laplace_em` as recommended by the developers. Additionally, we set the distributional form of the dynamics and emissions matrices to Gaussian.

The hyperparameters of dLDS primarily consists of the lagrange multipliers in the BPDN-DF objective including $\lambda_0, \lambda_1, \lambda_2$. We find the optimal value of these hyperparameters using a random search with a fixed budget of 1000 evaluations. For each hyperparameter, we uniformly sample over the log of the interval $[10^{-3}, 10^3]$ and evaluate it against the BPDN-DF objective. For the NASCAR experiment, we found that $\lambda_0 = 1.044, \lambda_1 = 0.254$, and $\lambda_2 = 0.023$ resulted in the best performance. For the Lorenz experiment, we found that $\lambda_0 = 0.628, \lambda_1 = 2.010$, and $\lambda_2 = 0.0124$ yielded the best performance.

For p-dLDS, the relevant hyperparameters consists of the SBL-DF dynamics tradeoff $\xi$, and the offset window size $S$. We use random search with a budget of 1000 samples to determine the values of $S$ and $\xi$ and fit a separate model for each set of hyperparameters. In the NASCAR experiment, we isolate the effect of the probabilistic inference procedure by setting $S = T$, removing the influence of the time-varying offset term. For $\xi$, we perform a random search by uniformly sample over the log of the interval $[10^{-3}, 10^3]$ and found that $\xi = 0.945$ was optimal. For the Lorenz experiment, we also optimize for the window size $S$ by uniformly sample a discrete index on the interval $\{2, \ldots, T\}$.

For the Lorenz experiment, the optimized hyperparameters are $S = 85$ and $\xi = 8.928$. For the real dataset, the optimal offset is $S = 76$ which is smaller than the timescale of p-dLDS coefficient switching (around 150 time points), suggesting that the same DO dynamics may persist even as the fixed points of the system fluctuates throughout the experiment.

### F.2 Hardware Specification

We perform hyperparameter sweep on our institution's HPC cluster using small-scale CPU resources which consists of Dual Intel Xeon Gold 6226 CPUs. Once hyperparameters have been optimized, it is possible to run each experiment within approximately 2 hours on the 2020 edition of the M1 Macbook Pro.

## G Description of Clinical Neurophysiology Data

Data was collected as part of a study investigating deep brain stimulation for treatment-resistant depression (TRD). The study is pre-registered in ClinicalTrials.gov (identifier NCT04106466). The study protocol was approved by the IRB (identifier IRB00066843). Informed consent was obtained from participants before participation in the trial. Patients receive no monetary compensation, but instead have their DBS electrodes and Summit RC+S IPG device provided free of charge. The analysis focused on LFP signals from a single participant with all personally identifiable information removed.

## H Limitations

While our proposed method demonstrates strong performance in our experiments, there are many limitations. For instance, our approach does not have a strong mechanism for generating future unseen coefficients. Our assumed coefficient transition model is primarily motivated by our desire to obtain smooth coefficients over time. However, we believe that they may be more complex transition models that can both capture persistent activity in challenging systems while also being an accurate forecaster, such as a deep learning based transition model. Another limitation of our approach is that our method assumes smoothness in the latent space. However, we do not explore the possibility of having sparse structure in the latent space which can be easily accomplished in BPDN-DF by adding an L1 penalty over $x$.

