# OpenReview forum: "Probabilistic Decomposed Linear Dynamical Systems for Robust Discovery of Latent Neural Dynamics"
_NeurIPS.cc/2024/Conference — NeurIPS 2024 poster_

### Official Review · Reviewer_mAN7 · 2024-06-27

**Soundness:** 4
**Presentation:** 4
**Contribution:** 3
**Rating:** 7
**Confidence:** 5

**Summary:**

This paper develops a novel model of latent neural dynamics that builds on existing work in switching and time-varying linear state space models. In particular, the authors propose a full probabilistic formulation of decomposed LDS models and provide a variational EM algorithm to do inference in these models. They apply their method to various synthetic and real neural datasets, showing improved performance over the primary time-varying linear alternative methods.

**Strengths:**

Overall, the proposed model and method are original and are a positive contribution to the modeling of neural dynamics. The authors correctly point out shortcomings of both the SLDS and dLDS approaches, and develop a new model that addresses shortcomings. In particular, the probabilistic formulation of dLDS dramatically improves the models performance, and the flexibility given by the time-varying linear combination of DOs helps p-dLDS perform well in situations that the SLDS is not well suited for. The evaluation is high-quality, as the paper compares relevant methods in four different tasks. Finally, the approach is well-motivated, and more often than not the methods and model are clearly described.

**Weaknesses:**

The model and fitting method have two added complexities that may make it harder than needed to reliably fit the model parameters and make inferences about the data. In particular, the update step sparsity coefficients requires a multi-step approximate procedure and the time-varying offset term depends on a hyperparameter that can take on a wide number of values. These points are discussed in more detail in the questions.

**Questions:**

- The approach for determining `b_t` via a moving average accomplishes the goals of learning a changing fixed point. However, it appears to be suboptimal for cases where the fixed point can change both rapidly or slowly (i.e. across different timescales) because of the fixed window size. Additionally, the requirement to search over a wide number of window sizes pose schallenges for optimization. I'd encourage the authors to consider other approaches. Could the bias term also be determined via a sum over a dictionary with shared coefficients `c`?

- Relatedly, can the authors provide more details of the hyperparameter selection scheme for the window size? The paper states that for the Lorenz experiment, the window size was sampled uniformly from 2 to the length of the time series. Does the procedure follow the dLDS hyperparameter selection, in which 1000 random choices of `S` and `\xi` were generated and a separate model was fit given each setting of hyperparameters? What was the optimal offset values on the real datasets, and does this relate to the timescales of the p-dLDS coefficient switching?

- The parameterization and update step for the sparsity coefficients appears to work well on the simulated and real datasets. However, it does require a multistep approximate procedure for the update. I'm wondering if the authors have considered simplifications to the model that may allow for using simpler inference in this step while still achieving the goals of learning sparse coefficients? For example, does it work to model each $\gamma_t$ independently? This appears to be very similar to the original SBL-DF algorithm and the first step of the sparsity coefficient update which initializes $q(c) q(\gamma)$ using SBL-DF. How much improved performance does the proposed model & methods have relative to this baseline?

- In the reaching experiment, the reach angles are classified using the discrete states or p-dLDS coefficients and not the continuous states. However, I would expect classification using the continuous states to outperform this in both cases. I would suggest the authors further justify that comparison in the text and compare to classification based on the continuous states.

Minor comments

- It appears that eq 5 describes the joint distribution $p(c_t, \gamma_t | c_{t-1})$, not just $p(c_t \mid c_{t-1})$.

**Limitations:**

The authors address the limitations in the text.

---

> ### Author Rebuttal · Authors · 2024-08-07
>
> We thank Reviewer mAN7 for their thoughtful feedback and suggested improvements to our experiments. We are encouraged that they found our work to be "high-quality" and a "positive contribution to the modeling of neural dynamics".
>
> **W1:** While p-dLDS does require more complex mathematical concepts, the main goal of this work is to develop a procedure that is easier and more reliable to work with than dLDS. Specifically, p-dLDS actually simplifies the fitting process by reducing the total number of hyperparameters for inference to two: the SBL-DF tradeoff $\xi$ and the offset window size $S$. In contrast, dLDS needs three hyperparameters ($\lambda_0$, $\lambda_1$, and $\lambda_2$) for determining the tradeoff between dynamic reconstruction, coefficient sparsity, and coefficient smoothness. In practice, manually balancing these tradeoffs between each of the Lagrange multipliers can be difficult and time-consuming. p-dLDS eliminates the need for manual tuning of these Lagrange multipliers through probabilistic inference. We estimate $\lambda_0$ with the inferred covariance $\Sigma_x$, $\lambda_1$ with $\gamma_{t,k}$'s from SBL-DF, and $\lambda_2$ with $\sigma^2_{t-1,k}$ also from SBL-DF. In practice, we find that our model is much easier to tune compared to the original dLDS model. As demonstrated in the results presented here, the outcomes are more accurate and robust for p-dLDS than dLDS.
>
> **Q1:** We agree that our approach for the offset may be limited when the fixed point changes across different timescales. We did consider a cost-based dictionary learning approach for the offset term early on in our work, but found it challenging to prevent the learned dynamics from collapsing to the degenerate scenario described in Lemma 1. Often, our offset dictionary would span the same subspace as the direction of the dynamics, preventing the learning of meaningful structure in the DOs.
> Moreover, this approach introduces additional complexity with multiple hyperparameters (e.g. for dictionary size, coefficient structure, and more) and requires an expensive iterative solver to infer offset coefficients, further slowing down training. In contrast, our moving average approach is simple, requiring only a single hyperparameter (the window size), and can be efficiently computed in a single pass.
>
> **Q2:** Yes, the p-dLDS hyperparameter selection procedure is very similar to the dLDS one. We use random search with a budget of 1000 samples to determine the values of $S$ and $\xi$ and fit a separate model for each set of hyperparameters. In our search, we uniformly sample an integer from 2 to the length of the time series $T$. For the real dataset, the optimal offset is $S=76$ which is smaller than the timescale of the p-dLDS coefficient switching (around 150 time points). This suggests that the same DO dynamics may persist even as the fixed point of the system fluctuates throughout the experiment. We appreciate the reviewer's suggestion for clarification and will include these additional details about the hyperparameters in the supplementary materials and the camera-ready version if accepted.
>
> **Q3:** Great question! While we did not directly explore modeling $\gamma_t$'s independently, previous works [1,2,3] have shown that the introduction of a dynamically informed prior leads to lower error and faster convergence when compared to their independent counterparts in both probabilistic and cost-based inference approaches. In particular, Figures 8 and 10 in [1] demonstrates that SBL-DF (with dynamically informed $\gamma_t$'s) shows an improvement in error and convergence time over the static SBL (with independent $\gamma_t$'s). Additionally, we highlight that our inference procedure is computationally efficient, and significantly reduces the training time required for decomposed models (see additional pdf for runtime experiments).
>
> [1] O’Shaughnessy et al. "Sparse Bayesian learning with dynamic filtering for inference of time-varying sparse signals." 2019.
>
> [2] Charles et al. "Dynamic Filtering of Time-Varying Sparse Signals via $\ell _1 $ Minimization." 2016.
>
> [3] Charles et al. "Convergence of basis pursuit de-noising with dynamic filtering." 2014.
>
> **Q4:**  Thank you for your suggestions! We originally did not consider using the continuous latent state to classify because we wanted to show that the learned dynamics contained information about the reach angle. However, following your suggestion, we found that the classification accuracy based on the continuous latent state $x_t$ outperformed the classification accuracy from previous experiments in all models. Similar to our previous classification set up, features are generated from the continuous latent states by averaging $x_t$'s over all points in time. We highlight that classification based on the continuous states improved p-dLDS's top-1 accuracy to 57.60\% and top-3 accuracy to 94.87\% (see additional pdf).
>
> **Minor.** Thank you for your careful review! We will update these typos.

---

> > ### Comment · Reviewer_mAN7 · 2024-08-08
> >
> > Thank you for your thoughtful and informative responses to my questions and for exploring classifying based on the continuous state for the reach experiment. I think it be worthwhile to incorporate these points into the paper.
> >
> > I remain convinced that this paper should be accepted. I intend to keep my score the same, and I encourage the other reviewers who scored lower than I did to consider raising their score.

---

### Official Review · Reviewer_3hfN · 2024-07-07

**Soundness:** 3
**Presentation:** 3
**Contribution:** 2
**Rating:** 5
**Confidence:** 2

**Summary:**

The Authors propose a probabilistic extension to the "Decomposed Linear Dynamical Systems (dDLS)" method by Mudrik et al. (2024). The proposed p-dLDS belongs to a family of models specifically designed to describe neural activity from high-dimensional dynamical data. The effectiveness of p-dLDS is evaluated on a set of synthetic benchmarks and a real dataset.

As the primary use-case of p-dLDS -- modeling neural data -- is not my expertise, my evaluation capability for this paper is quite limited, as reflected in my confidence assessment.

**Strengths:**

On top of giving accurate predictions of future states, the proposed method recovers a sensible switching behavior for every one of the examples reported.

**Weaknesses:**

- The experimental results do not report the variance across independent runs. As most of the datasets are synthetic, these should be fairly straightforward to produce.
- Apart from adding a probabilistic structure to the sparse coefficients of dLDS, p-dLDS adds a slow-fast decomposition of the latent state (Eq. 4). Yet, no ablation study assessing the importance of this modeling choice is reported.

**Questions:**

- What is the running time of the different baselines? What is the computational complexity of p-dLDS?
- From the experimental evaluation, I can see that dLDS and p-dLDS behave quite differently. Have you observed any relationship between the posterior mean of the coefficients $\mathbf{c}_{t}$ in p-dLDS and the deterministic analog in dLDS?

**Limitations:**

N / A

---

> ### Author Rebuttal · Authors · 2024-08-07
>
> We thank Reviewer 3hfN for taking their time to review our work.
>
> **W1:** We report the standard deviation across independent runs in Tables 3, 4 and 5 in the supplementary materials. We omit these values in the main text due to limited space.
>
> **W2:** While we do not perform an ablation study for the offset term, we discuss theoretically in section 3.1 why the lack of an offset term in dLDS leads to a representation that does not generalize in systems that contain multiple fixed points. Namely, that any parameter setting in the original dLDS dynamics model reduces to a linear dynamical system (LDS) which is characterized by a single fixed point centered around the origin. The Lorenz experiment (section 4.1) illustrates how this limitation leads to a representation that does not align with the attractor lobes, inappropriately segments the system radially with respect to the origin (Fig. 2D), and leads to decreased performance in our quantitative metrics (Table 1). In contrast, we show that the offset term in p-dLDS alleviates these problems and enables the learning of a representation that aligns with the multiple fixed points of the Lorenz attractor.
>
>
> **Q1:** We include additional runtime experiments that sweep across a range of dictionary sizes and latent dimensions (see attached pdf). We observe that in general, p-dLDS is significantly faster than dLDS, and even matches rSLDS and SLDS in certain parameter settings. Our approach adopts the worst-case computational complexity of the SBL inference procedure which is $\mathcal{O}(TK^3)$ in time and $\mathcal{O}(TK^2)$ in memory due to a matrix inversion required to compute the posterior coefficient covariance [1]. However, as noted by [2], the introduction of an dynamically informed hyperprior reduces the number of overall iterations required for convergence and typically reduces the actual runtime by an order of magnitude.
>
>
> [1] Tipping. "Sparse Bayesian learning and the relevance vector machine." 2001.
>
> [2] O’Shaughnessy et al. "Sparse Bayesian learning with dynamic filtering for inference of time-varying sparse signals." 2019.
>
>
>
> **Q2:** In general, we observe that the posterior mean of the coefficients in p-dLDS will be different from the deterministic analog in dLDS. In Figures 2B, 2D, 4C and 4F, we illustrate these differences by plotting the inferred coefficients from both models side by side. We find that without the probabilistic inference procedure and the time-varying offset term, the inferred coefficients of dLDS can produce switching patterns that may not align with the ground truth (Figures 2B and 2D) and can oscillate unpredictably (Figures 4C and 4F).

---

### Official Review · Reviewer_M5AA · 2024-07-13

**Soundness:** 3
**Presentation:** 3
**Contribution:** 3
**Rating:** 7
**Confidence:** 3

**Summary:**

The paper proposes the probabilistic decomposed linear dynamical systems (p-dLDS) model, which extends the existing dLDS model. With the time-varying offset term and probabilistic formulation, p-dLDS improves upon the dLDS model in terms of robustness to noise and the ability to capture systems that orbit multiple fixed points. The authors show the advantage of the p-dLDS model over SLDS and dLDS through simulated and real datasets.

**Strengths:**

- Clarity: The paper is presented clearly, with easy-to-understand figures and descriptions for the model, inference procedure, and experimental results. The background and related work sections are also well-written, making the paper easy to understand for readers who are not experts in the field.

- Reproducibility: The paper includes code and hyperparameter settings to reproduce its experiments, which is crucial for the ML community.

- Significance: The probabilistic extension, in addition to the time-varying offset term, adds value to the recently developed dLDS, which suffers from dynamics noise.

**Weaknesses:**

- Experiments
Although existing experimental results are exciting, I have a couple of suggestions.

One is that, while the paper claims the robustness of p-dLDS to dynamics noise, it does not have experiments that show how robust p-dLDS is to noise. In other words, as you sweep the experiments with different dynamics noises, when do SLDS, dLDS, and p-dLDS show similar/different results? When does p-dLDS fail? Showing the advantage of p-dLDS over other models in a realistic setting of dynamics noise would be important.

In addition, the experiments set the number of DOs and discrete states the same for p-dLDS and rSLDS (e.g., Section 4.2). For a fair comparison, I think that the number of discrete states for rSLDS should be set separately on its own via cross-validation. It could be that rSLDS needs more discrete states to perform as well as p-dLDS on e.g., linear classification of reach directions.

- Interpretation of discrete states of p-dLDS
To my understanding, the paper does not explain how the learned p-dLDS parameters are used to segment the data to infer "discrete" states, as in Figures 2 and 4. In addition, for the NASCAR experiment, p-dLDS leads to fewer segmentations than four true segments. I am unsure whether we could say that this representation is more "parsimonious" while it seems like it leads to incorrect segmentations. In the Lorenz attractor experiment, it also seems like the rSLDS segmentation is more interpretable than p-dLDS segmentation.

**Questions:**

- When does the model break due to the inclusion of time-varying offset terms? In other words, are there cases or hyperparameter settings when the model fails? How should the users avoid this? Are the covariances in line 158 learned or user-specified?

- The paper resorts to using PCA to determine the dimensionality of the model. Is there a reason why the dimensionality is not chosen via cross-validation?

- Minor: In Section E.1 Figure 6 panel A, it seems like the ELBO peaks at the first few iterations, then drops, and finally converges to a value smaller than the peak. I wonder if there's an explanation for this.

- How many hyperparameters are there for p-dLDS? How are these chosen, and how simple or complex is the model selection process for p-dLDS compared to the hyperparameter sweep for rSLDS?

**Limitations:**

- As the authors noted, one limitation of p-dLDS is that it assumes Gaussian observation noise.

---

> ### Author Rebuttal · Authors · 2024-08-07
>
> Thank you for your careful reading of our submission and are encouraged that you find our work to be "well-written" and "adds value to the recently developed dLDS".
>
> **W1. Sweeping dynamics noise. Advantage of p-dLDS in a realistic setting.**
>
> We agree that it's important to understand our model's behavior under different noise conditions. In our experiments, we focus on covering a range of neurally plausible sources of noise and demonstrate that p-dLDS improves significantly over existing models on a variety of metrics. Moreover, we emphasize that our clinical neurophysiological experiment showcases exactly “the advantage of p-dLDS over other models in a realistic setting”. We use real data with real noise (e.g. noisy trial structure, unexpected head movements, etc.), and demonstrate that p-dLDS can capture meaningful structure where other models cannot.
>
> As discussed in our introduction, dynamical noise can arise from numerous sources, making it challenging to thoroughly explore all scenarios within the scope of this paper.  Without a specific end application in mind, it is difficult to construct a experiment that fully addresses this point.
>
>
> **W2. Selecting number of discrete states for rSLDS via cross validation.**
>
> Great point! In our original experiments, we focus on approximately matching the model parameters to control for the model complexity. However, we include an additional experiment where we sweep the number of discrete states for rSLDS (see additional pdf). While more discrete states did improve rSLDS performance, it still underperforms with respect to p-dLDS at a lower parameter count, highlighting the inherent limitations of using a switched formulation to model a continuum of signals.
>
> **W3. How are "discrete" states defined for pdLDS? Parsimonious and interpretable segmentations.**
>
> As noted in the captions of Figures 2B and 2D, “discrete” states for dLDS models are colored according to the “dominant coefficient” which we define as the dynamic operator (DO) state with the largest coefficient magnitude. We plan on making our definition more clear in the main text in the camera-ready version of the paper, if accepted.
>
> For NASCAR, we respectfully disagree that pdLDS "leads to incorrect segmentations". Due to the pdLDS model formulation, the learned model components are capturing the notion that different track segments can have their movement captured by the same DO (i.e., using the same movement but backward). While this results in fewer segments than the ground truth switched model the data was generated with, it correctly identifies redundancies in the switched segments that may be important to capture in many applications. The ability to identify these redundancies is a key advantage of pdLDS over switched models. While you certainly may want to distinguish these cases in a specific application, this is easily done by looking at the sign of the coefficients. We emphasize here again that pdLDS recovers the track segments despite noise (unlike other methods), leading to a variety of quantitative performance improvements.
>
>
> In the Lorenz experiment, while rSLDS segmentations may seem simpler and more interpretable, it inappropriately under-segments the true system by assigning the same state to both fast and slow segments, which loses important dynamics information about speed. This behavior can be problematic for signals that vary across a continuum, such as those from the reaching experiment. Moreover, our quantitative assessment (Table 1) shows that p-dLDS's representation leads to improved performance across various metrics, supporting its value despite apparent complexity.
>
> **Q1:** While we agree that understanding the limitations of the time-varying offset term is important, fully characterizing the model's failure modes without a specific application in mind is challenging. There are numerous ways to study when the model breaks due to the offset terms, and the most relevant ones depend on the specific use case.
>
> Regarding the covariances, lemma 2 allows users to implicitly specify the sum of both covariances $\Sigma_l + \Sigma_b$ through the selection of the smoothing window hyperparameter $S$. Given a particular $S$, our fitting procedure estimates the value of the covariances $\Sigma_l + \Sigma_b$.
>
> **Q2:** We use PCA to determine the size of the latent space due to its simplicity, computational efficiency, and ability to control for parameter count across different models. In computational neuroscience, PCA is a widely used and well established method that has provided numerous scientific insights [citations available upon request]. In contrast, cross-validation can be costly and may select different latent space sizes for different models, which can confound differences due to parameter count with differences in the latent dynamics model. By selecting the latent dimension across all models using PCA, we control for model complexity and can more directly compare their learned representations.
>
> **Q3:**  We believe this results from the training dynamics introduced by using SGD and Momentum, combined with the challenging non-convex optimization landscape defined by our ELBO objective. Generally speaking, variational inference is only guaranteed to converge to a local optimum.
>
> **Q4:** Hyperparameters (HPs) include $M$, $N$, and $K$ for model dimensionality; $\xi$ and $S$ for coefficient inference; and any SGD HPs (e.g., learning rate, momentum). In general, HPs can be estimated using either domain knowledge or cross-validation. While p-dLDS has a more complex parameter sweep than rSLDS, it is easier to fit than dLDS since it does not require the manual balancing of the BPDN-DF Lagrange multipliers. This greatly simplifies the fitting process for decomposed models.

---

> ### Comment · Reviewer_M5AA · 2024-08-09
>
> I would like to thank the authors for their response to my comments and clarifications. I would like to raise my score from 6 (weak accept) to 7 (accept).

---

### Official Review · Reviewer_jjeY · 2024-07-15

**Soundness:** 3
**Presentation:** 4
**Contribution:** 2
**Rating:** 7
**Confidence:** 4

**Summary:**

The paper presents a probabilistic version of the dLDS model, first presented by Mudrik et al (2024). The primary impetus for this model was to present a version of the dLDS that was more robust to noise, although the authors here also included a slowly-evolving offset term meant to capture evolving fixed points. Inference is conducted by a variational objective which is optimized with a combination of ad-hoc but reasonable methods. The paper paves the way for many other probabilistic state-space models including those with non-Gaussian measurements.

**Strengths:**

Originality - The model is original in the sense that it has not been presented previously.
Quality - The paper is mostly well written and well reasoned. I see virtually no major conceptual flaws and only minor problems. Both the simulation experiments and analysis demonstration are well conceived and of interest to the computational neuroscience community. The model is well conceived and represents a natural extension of the dLDS model.
Clarity - Very well organized, easy to follow, and a pleasure to read.
Significance - I wouldn’t exactly call this a weakness but the extension to dLDS model is so natural as to be virtually obvious. The contribution itself, while seemingly inevitable, was nonetheless executed by the authors first (props to them) and with high quality. It serves as a strong rung in what is sure to be a dLDS ladder.

**Weaknesses:**

The primary weakness is in some technical confusion that i have the with paper. I detail below.

**Questions:**

I am a bit confused about the decomposition of the coefficient prior defined on lines 168-169 following equation (5). Here the authors claim p(c_t|c_t-1, \gamma_t) is defined by a factorization N(c_t-1,\sigma_t-1)N(0,\gamma_t). The authors state that the first of these factors encourages smoothness while the second encourages sparsity. However, it is not at all clear to me that this makes sense. First, the prior is a conditional distribution over c_t for a single coefficient, so what is the factorization over exactly? Did the authors mean to present a joint distribution? Over which variables? I can see that the authors would like c_t to have both properties (sparsity, smoothness) but this particular specification doesn’t make sense. Second, \sigma_t and its inference is undefined anywhere in the paper. Third, on lines 215-216 the authors state that parameters can be learned by SGD “which is possible when we assume that the covariance matrices have diagonal structure.” Do the authors specifically mean the posterior covariances (eq 8), the prior covariances over the latent variables (line 158)? Either way, what is the limitation on SGD learning the parameters when these covariance matrices are not diagonal? Lastly, the authors state on line 187 “the approximate posterior becomes…[some Gaussians]”. It is not clear from the text if the Gaussian form of this variational posterior is a choice of the authors or if this is a variational optimum that drops out of the model structure and factorized structure of q. Can the authors please clarify?

**Limitations:**

No major conceptual limitations but technical details should be clarified

---

> ### Author Rebuttal · Authors · 2024-08-07
>
> Thank you for taking the time to review our paper thoroughly. We are encouraged that you find our work to be "original", "a pleasure to read", "well-conceived", and of "interest to the computational neuroscience community".
>
> **Q1 and Q2: Factorization of the coefficient transition and Definition of $\sigma_t$**
>
> We clarify that the transition density $p(c_{t,k} | c_{t-1,k}, \gamma_{t,k} )$ over the coefficients $c_t$ has the following functional form:
>
> $$
> \begin{align}
>     p(c_{t,k} | c_{t-1, k} , \gamma_t) &\propto \exp \left(- \frac{c_{t,k}^2 }{2\gamma_{t,k}} - \frac{(c_{t,k} - c_{t-1,k})^2} {2 \sigma_{t-1,k}^2}\right) \\\\
>     &= \exp \left(- \frac{c_{t,k}^2 }{2\gamma_{t,k}} \right) \exp \left( - \frac{(c_{t,k}- c_{t-1,k})^2} {2 \sigma_{t-1,k}^2}\right) \\\\
>     &\propto N(c_{t,k} ; 0, \gamma_{t,k}) N(c_{t,k} ; \hat{c}\_{t-1,k}, \sigma_{t-1,k}^2)
> \end{align}
> $$
>
> In the first line, we propose a density that captures the constraints of sparsity and smoothness for the inferred coefficients $c_{t,k}$. A small  variance around 0, $\gamma_{t,k}$, promotes sparsity while a small variance around the previous coefficient, $\sigma^2_{t-1,k}$, promotes smoothness. The second line factorizes this density into two separate terms and the third line shows that each term is proportional to a normal distribution with their respective parameters.
>
> While the idea of combining two shrinkage effects in a single density has been explored in previous works [1,2,3,4], those approaches generally require manual balancing of the two penalties. In contrast, we estimate these parameters using SBL-DF results. Specifically, SBL-DF produces estimates for the sparsity variance $\gamma_{t,k}$ through the estimated hyperparameter posterior $q(\gamma_{t,k} )$ and the smoothness variance $\sigma^2_{t-1,k}$ through the approximate coefficient posterior $q(c_{t,k} ) = N(c_{t,k} ; \hat{c}\_{t,k} ,\sigma_{t,k}^2 )$.
>
> We thank the reviewer for pointing out that our discussion on the coefficient transition densities could be improved and will incorporate this discussion into the camera-ready version if accepted.
>
> [1] Irie, Kaoru. "Bayesian dynamic fused LASSO." 2019.
>
> [2] Casella et al. "Penalized regression, standard errors, and Bayesian lassos." 2010.
>
> [3] Li and Lin. "The Bayesian elastic net." 2010.
>
> [4] Kakikawa et al. "Bayesian fused lasso modeling via horseshoe prior." 2023.
>
> **Q3: SGD over covariance matrices.**
>
> We clarify that we assume a diagonal structure for the posterior covariances. Estimating general covariance matrices $\Sigma$ is challenging both mathematically and computationally due to the requirements for a valid $\Sigma$ (e.g., symmetry, positive semidefiniteness). Vanilla SGD over individual matrix entries does not inherently respect these constraints and does not guarantee that the solution is a valid covariance matrix. Typically, solving for $\Sigma$ involves a semidefinite program (SDP) requiring specialized methods like ellipsoid or interior point solvers [1]. However, we simplify this problem by considering only when $\Sigma$ is diagonal (i.e., $\Sigma = {\rm diag} (\sigma_1 ,\dots, \sigma_n)$ ). This allows us to simplify general SDP optimization to an unconstrained problem by operating in the log space of the diagonal variance terms. We find that this approach works well in our experiments and is computationally efficient (see additional pdf for runtimes).
>
>  [1] Vandenberghe and Boyd. ``Semidefinite programming.'' SIAM review 1996.
>
> **Q4: Derivation of the approximate posterior.**
>
> Thank you for pointing out that equation 8 (following line 187) could be improved in clarity. It is the variational optimum that results from the model structure assumed in line 157. We derive this result below. First, we acknowledge two typos:
>
> 1. The joint distribution of pdLDS (equation 6) is missing emission densities and will be updated to,
>
> $$ \begin{align}
> p(x, y,c, \gamma | \theta ) = p(x_1) \left[ \prod_{t=1}^ T p(y_t | x_t) \right] \left[ \prod_{t=1} ^ {T-1} p(x_{t+1} | x_{t}, c_{t})\left[ \prod_{k=1}^{T} p(c_{t+1,k} | c_{t,k}, \gamma_{t+1,k}) p (\gamma_{t+1, k} | c_{t,k}) \right] \right] \tag{1}
> \end{align}$$
>
> 2. The formula for the optimal coordinate ascent variational update (equation 7) should be the exponentiated log of the joint distribution [1] and will be update to,
>
> $$
> \begin{align}
> q(x) \propto \exp \left( \mathbb{E}_{q(c, \gamma)} \left[ \log p(x, y, c, \gamma | \theta)  \right] \right). \tag{2}
> \end{align}$$
>
> We proceed by substituting equation 1 above into equation 2, dropping out constant terms,
>
> $$\\mathbb{E}\_{q(c, \\gamma)} \\left[ \\log p(x, y, c, \\gamma | \\theta)  \\right] = \\mathbb{E}\_{q(c, \\gamma)} \\left[   \\log p(x_1) + \sum_{t=1}^ T \\log p(y_t | x_t)  + \\sum_{t=1} ^ {T-1} \\log p(x_{t+1} | x_{t}, c_{t}) \\right] + {\\rm const.} \tag{3}$$
>
>
> Next, our assumed decomposition in line 157 gives us that $p(x_{t+1} | x_{t}, c_t) = p(x_{t+1} = l_{t+1} + b_{t+1})$. The distribution over $x_{t+1}$ is computed as,
>
> $$
> \begin{align}
>     p(x_{t+1}=l_{t+1} + b_{t+1}) &= p(l_{t+1} | l_t, c_t) \star p(b_{t+1} | b_{t})  \\\\
>     &= N(x_{t+1} ; l_t + F_t l_t + b_t, \Sigma_x) \tag{5}
> \end{align}
> $$
>
> where we have defined the convolution operator $\star$ and $\Sigma_x =\Sigma_b + \Sigma_l$. Substituting the above equations 5 and 3 into the above equation 2 gives us the optimal variational update,
>
> $$q(x) \propto N(\mu_1, \Sigma_1) \prod_{t=1}^T N(y_t; D x_t, \Sigma_y)  \prod_{t=1}^{T-1} N(x_{t+1} ; l_t + \hat{F}_t l_t + b_t, \Sigma_x)
> $$
>
> where $\hat{F}\_t$ is estimated from samples from $\hat{c}\_{t} \sim q(c,\gamma)$. We note that the additional emissions term in the above equation 1 leads to an additional term in our above equation 8.
>
> Again, we thank the reviewer for bringing this to our attention. We plan on correcting the typos and will include this discussion into the camera-ready version if accepted.
>
> [1] Blei et al. "Variational inference: A review for statisticians." 2017.

---

### Author Rebuttal · Authors · 2024-08-07

Thank you to all four reviewers for their insightful feedback. We respond to reviewers individually below, but provide a description of our additional experiments here. Please see our attached pdf for figures relevant to the experiments below.

**Experiments:**

1. **Increasing the number of rSLDS discrete states in the Reach Experiment:** Reviewer M5AA suggests that the optimal number of rSLDS discrete states may differ from the one used. We sweep the number of discrete states $K$ from 4 to 20 with a step size of 2 and compare against the p-dLDS results at $K=4$. We find that although increasing the number of discrete states leads to improvements in rSLDS performance, it does not outperform p-dLDS performance at $K=4$.


2. **Training Runtime:** Reviewer 3hfN recommends that we compare the runtimes across different models. We report the time per training iteration when sweeping across the dictionary size $K$ and the latent dimensions $N$ separately from 5 to 50 with a step size of 5. For all experiments, we report training times on a single time series with length $T=1000$ and observation dimension $M=100$. When sweeping $K$, we fix $N=10$. Similarly when sweeping $N$, we fix $K=10$. We see that the p-dLDS inference procedure significantly reduces the training time required for decomposed models.


3. **Reach Classification using Continuous Latent States:** Following Reviewer mAN7's suggestion, we compared classification accuracy based on the dynamics ($z$ or $c$) to that based on the continuous states ($x$). We computed classifier features from $x_{1:T}$ by taking the average over time for each trajectory, similar to section D.5. This improved the Top-1 and Top-3 accuracies for all models, indicating that location in latent state space can be highly informative about specific reach directions. In this experiment, p-dLDS achieved Top-1 and Top-3 accuracies of 57.69\% and 94.87\% respectively.

---

### Decision · Program_Chairs · 2024-09-25

**Decision:**

Accept (poster)

**Comment:**

This is a nice extension of recent work on decomposed LDS models. The novel contribution is a probabilistic formulation of the model and a variational EM framework for state and parameter estimation. I was particularly impressed with the improvements in state segmentation on real datasets. Please incorporate the changes and clarifications described during the discussion period. I think this will be a solid contribution to NeurIPS.